*Method*

# scClassify: sample size estimation and multiscale classification of cells using single and multiple reference

Yingxin Lin[1,2], Yue Cao[1,2], Hani Jieun Kim[1,2,3], Agus Salim[4,5,6], Terence P Speed[6] ID, David M Lin[7], Pengyi Yang[1,2,3,*] ID & Jean Yee Hwa Yang[1,2,**]

## Abstract

Automated cell type identification is a key computational challenge in single-cell RNA-sequencing (scRNA-seq) data. To capitalise on the large collection of well-annotated scRNA-seq datasets, we developed scClassify, a multiscale classification framework based on ensemble learning and cell type hierarchies constructed from single or multiple annotated datasets as references. scClassify enables the estimation of sample size required for accurate classification of cell types in a cell type hierarchy and allows joint classification of cells when multiple references are available. We show that scClassify consistently performs better than other supervised cell type classification methods across 114 pairs of reference and testing data, representing a diverse combination of sizes, technologies and levels of complexity, and further demonstrate the unique components of scClassify through simulations and compendia of experimental datasets. Finally, we demonstrate the scalability of scClassify on large single-cell atlases and highlight a novel application of identifying subpopulations of cells from the Tabula Muris data that were unidentified in the original publication. Together, scClassify represents state-of-the-art methodology in automated cell type identification from scRNA-seq data.

**Keywords** cell type hierarchy; cell type identification; multiscale classification; sample size estimation; single-cell

**Subject Categories** Chromatin, Transcription & Genomics; Computational Biology; Methods & Resources

**Mol Syst Biol. (2020) 16: e9389**

## Introduction

Cell type identification is an essential task in single-cell RNA-sequencing (scRNA-seq) data analysis (Trapnell, 2015). The most common approach to identifying cell types within scRNA-seq data is unsupervised clustering (Kiselev *et al*, 2017; Wang *et al*, 2017; Freytag *et al*, 2018; Diaz-Mejia *et al*, 2019) followed by manual annotation based on a set of known marker genes (Kolodziejczyk *et al*, 2015). However, the number of clusters is rarely known in advance, and the annotation of clusters is subjective, time-consuming and highly dependent on prior knowledge of previously identified marker genes (Grün & van Oudenaarden, 2015). This can introduce bias in the analysis towards the better characterised cell types. With the increasing availability of well-annotated scRNA-seq datasets, an alternative approach is to train supervised learning methods on reference scRNA-seq datasets with high-quality annotation to classify cells in new/query datasets (Abdelaal *et al*, 2019). Compared to unsupervised clustering, supervised learning methods can automate the cell type identification process and reduce the bias associated with marker gene selection in cell type annotation (Zhao *et al*, 2019).

While many major cell types can be divided into subtypes in a hierarchical fashion, forming what we call a "cell type hierarchy" (Bakken *et al*, 2017), current supervised learning methods typically classify cells directly to a "terminal" cell type and ignore any hierarchical relationships between cell types. Such a "one-step" classification approach does not take into account the number of cells in the reference dataset (i.e. sample size) that is needed to train a model for classifying query cells to the "terminal" cell types. Additionally, these "one-step" classification models typically use a single reference dataset, and if a certain cell type in a query dataset does not exist in the reference dataset, it will be forcibly assigned to an unrelated cell type. These limitations collectively contribute to

---

1 School of Mathematics and Statistics, University of Sydney, Sydney, NSW, Australia
2 Charles Perkins Centre, University of Sydney, Sydney, NSW, Australia
3 Computational Systems Biology Group, Children's Medical Research Institute, University of Sydney, Westmead, NSW, Australia
4 Department of Mathematics and Statistics, La Trobe University, Bundoora, VIC, Australia
5 Baker Heart and Diabetes Institute, Melbourne, VIC, Australia
6 Bioinformatics Division, Walter and Eliza Hall Institute of Medical Research, Parkville, VIC, Australia
7 Department of Biomedical Sciences, Cornell University, Ithaca, NY, USA
 *Corresponding author. Tel: +61 2 9351 4534; E-mail: pengyi.yang@sydney.edu.au
 **Corresponding author. Tel: +61 2 9351 3012; E-mail: jean.yang@sydney.edu.au

misclassification, which could be avoidable by estimating the sample size required for model training, accounting for hierarchical relationships between cell types and including multiple closely related reference datasets where available.

To address these challenges, we developed scClassify, a multi-scale classification framework based on ensemble learning for

accurate cell type identification (Fig 1A). scClassify first constructs a cell type tree from a reference dataset where cell types are organised in a hierarchy with increasingly fine-tuned annotation, using a log-transformed size factor-normalised expression matrix as an input. Next, scClassify uses a combination of gene selection methods and similarity metrics to develop an ensemble of classifiers

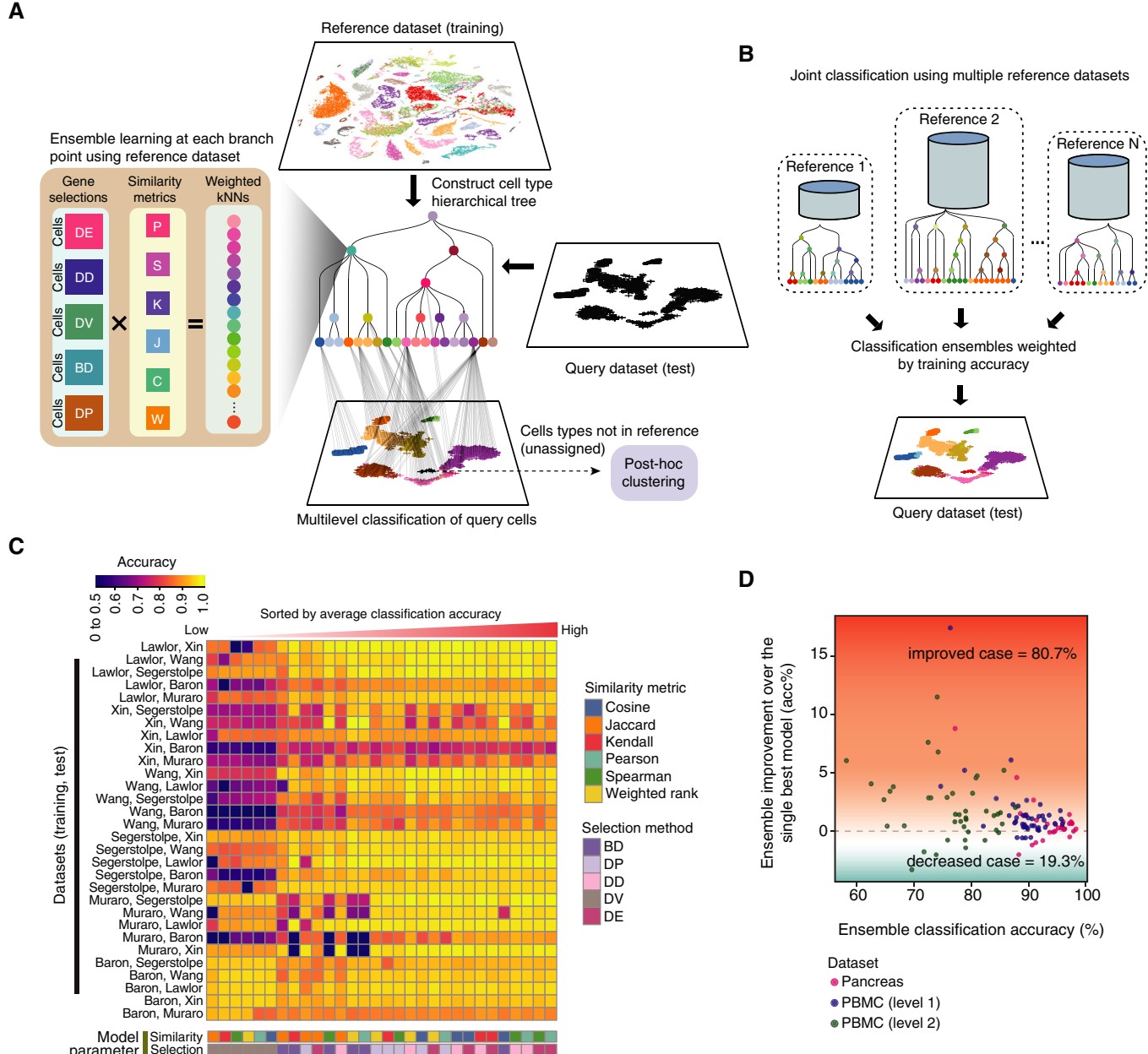

**Figure 1. scClassify framework and ensemble model construction (see also Fig EV1).**

A  Schematic illustration of the scClassify framework. Gene selections: DE, differentially expressed; DD, differentially distributed; DV, differentially variable; BD, bimodally distributed; DP, differentially expressed proportions. Similarity metrics: P, Pearson's correlation; S, Spearman's correlation; K, Kendall's correlation; J, Jaccard distance; C, cosine distance; W, weighted rank correlation.

B  Schematic illustration of the joint classification using multiple reference datasets.

C  Classification accuracy of all pairs of reference and test datasets was calculated using all combinations of six similarity metrics and five gene selection methods.

D  Improvement in classification accuracy after applying an ensemble learning model over the best single model (i.e. weighted $k$NN + Pearson + DE).

that capture cell type characteristics at each non-terminal branch node of the cell type hierarchy. These classifiers are then integrated to make predictions for every cell at each branch node of the cell type hierarchy. Depending on the sample size of each cell type in the reference dataset, scClassify may assign a cell from a query dataset to a non-terminal intermediate cell type and not classify it any further in the hierarchy. To account for the possibility that the query cell types may not be present in the reference dataset, scClassify allows cells from the query dataset to be labelled as "unassigned" (Figs 1A and EV1A). scClassify also enables the estimation of the number of cells required for accurately discriminating between cell types and subtypes anywhere in the cell type hierarchy. This is an important component for the experimental design and generation of reference datasets with sufficiently large numbers of cells, which is critical for nuanced identification of cell types. For unassigned cells, scClassify uses a *post hoc* clustering procedure for novel cell type discovery (Fig 1A), and when multiple reference datasets are available, scClassify takes advantage of this by enabling joint classification of cells in a new dataset by using multiple references (Fig 1B). The joint classification procedure increases the sample size for model training, improves cell type classification accuracy and reduces the number of unassigned cells.

## Results

### scClassify benefits from ensemble learning and outperforms existing supervised methods

We demonstrate the value of ensemble learning with a collection of seven PBMC datasets (Ding *et al*, 2020; Data ref: Ding *et al*, 2020) generated by different protocols and six publicly available human pancreas scRNA-seq datasets (Fig EV1B). We first evaluated the performance of 30 individual classifiers on the pancreas data collection by training each on one dataset and testing it on another, with a weighted *k*-nearest neighbour (*k*NN) classifier using one of five gene selection methods and one of six similarity metrics (Fig 1C and Dataset EV1). The heatmap (Fig 1C) highlights the diversity in performance across different parameter settings (with average accuracy ranging from 72 to 93%), suggesting that different parameter combinations capture different cell type characteristics. While the differential expression (DE) gene selection method was the best single classifier, followed by the weighted *k*NN with Pearson's similarity metric, we found that the ensemble of weighted *k*NN classifiers trained by all 30 combinations of gene selection methods and similarity metrics led, in most cases, to a classification accuracy higher than that achieved by the single best model (Fig 1D). The ensemble classifier was therefore used in all benchmarking.

We compared the performance of scClassify against 14 other single-cell-specific supervised learning methods (Kiselev & Yiu, 2018; Lieberman *et al*, 2018; Lopez *et al*, 2018; preprint: Wagner & Yanai, 2018; Alquicira-Hernandez *et al*, 2019; Aran *et al*, 2019; preprint: Boufea *et al*, 2019; de Kanter *et al*, 2019; Pliner *et al*, 2019; Tan & Cahan, 2019; Ma & Pellegrini, 2020) (Appendix Table S1). The $6 \times 5 = 30$ (training and test) pairs from the pancreas data collection of six studies came in two groups: easy cases ($n = 16$),

where all cell types in the test data are found in the training data, and hard cases ($n = 14$), where the test data contained one or more cell types not present in the training data. The results are summarised in Figs 2A and EV2A, and an example is shown in Appendix Fig S1. On average, scClassify achieved a higher accuracy than the other methods, with the difference being greater among the 14 hard cases than the 16 easy ones. For the collection of six PBMC datasets, we evaluated scClassify at two levels of the cell type hierarchy, coarse ("level 1") or fine ("level 2"), each leading to $7 \times 6 = 42$ (training and test) pairs. We found that scClassify effectively annotates the cell types and produces higher accuracy rates in most cases (Figs 2B and EV2A), with the improvement being greater at level 2 than at level 1. Results for all pairwise comparisons of methods are summarised in Fig 2C. In terms of computational efficiency and memory required, we trained and tested each method on classifying the Tabula Muris dataset (The Tabula Muris Consortium, 2018; Data ref: The Tabula Muris Consortium, 2018) with varying numbers of cells or cell types and found that scClassify is comparable to other existing methods and can be applied to classify datasets with a large number of cells (Appendix Fig S2).

To evaluate the robustness and stability of scClassify on cell type classification, we repeatedly resampled each training dataset in the pancreas data collection and performed scClassify on the total of 30 training and test pairs (see Materials and Methods). We found that the classification accuracy of scClassify using a subset of training data is highly reproducible and highly concordant with the results from the full training dataset (Fig EV2B). To assess the impact of hyperparameters on scClassify, we used the 30 training and test pairs from the pancreas data collection and evaluated three key hyperparameters (see Materials and Methods). We illustrate in Fig EV2C that the choice of k considered in weighted *k*NN had minimal impact on the performance of scClassify. We found that the dynamic and pre-defined threshold in correlation threshold determination for cell type classification are highly concordant in the 16 easy cases, while in the hard cases, the dynamic threshold is generally better than a hard-coded threshold (Fig EV2D). Finally, high consistency of performance was observed from using a different maximum number of children per branch node of the HOPACH tree (Appendix Fig S3).

### Sample size estimation of reference datasets

To facilitate a complete design and identification framework, scClassify enables estimation of the number of cells required in a reference dataset to accurately discriminate between any two cell types at a given level in a given cell type hierarchy. It does so by fitting an inverse power law (Mukherjee *et al*, 2003) (see Materials and Methods). The procedure requires no assumption on the distribution of the training dataset or the accuracy. We expect that the accuracy of cell type classification will increase with increasing sample size and converge to a maximum. To evaluate this approach, for a given accuracy and the corresponding sample size from the learning curve estimated from the pilot data (Fig 3A red line, Fig EV3A), we performed an *in silico* experiment by randomly selecting samples of cells of different sizes from the full reference dataset and built a cell type prediction model. Finally, the model was validated on an independent set of cells, and the corresponding *in silico* experiment accuracy

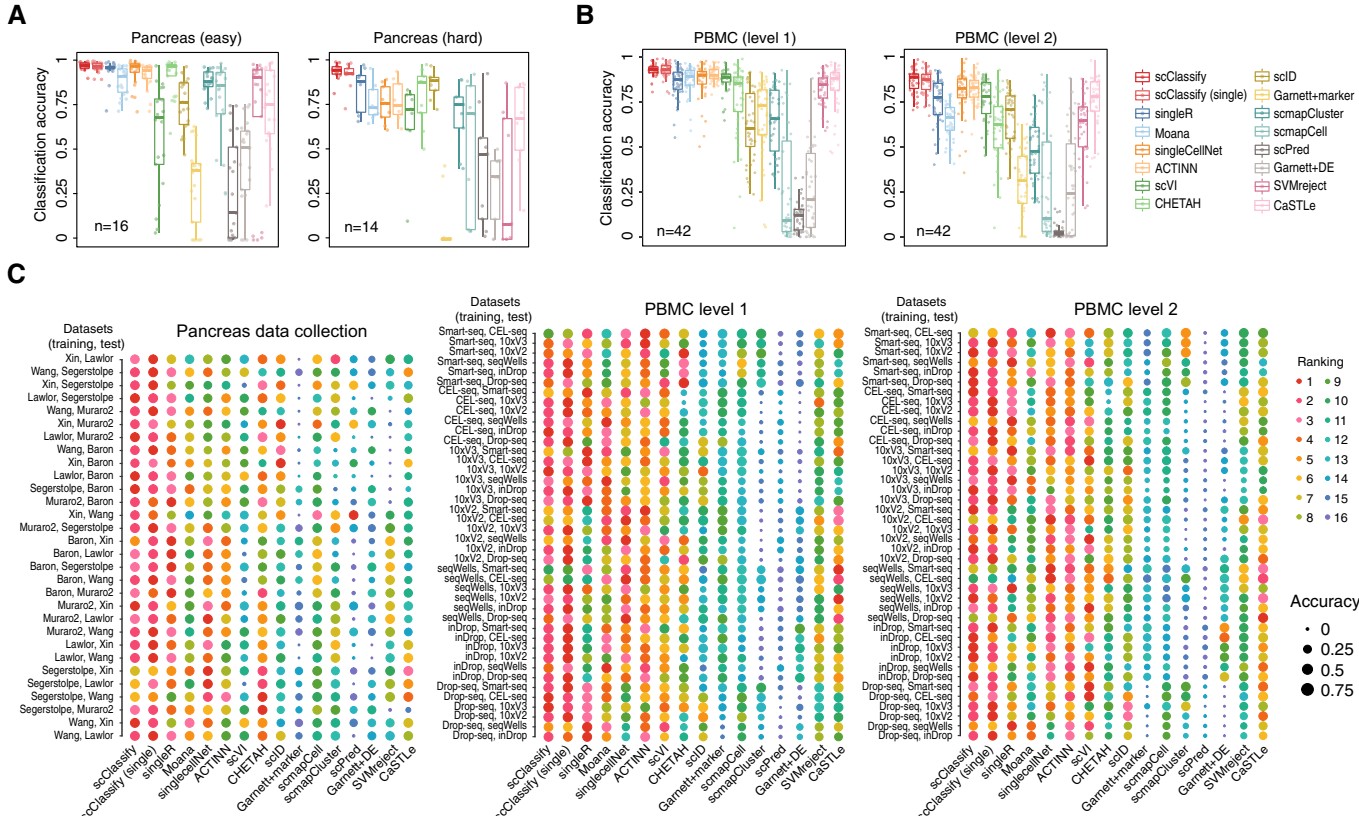

**Figure 2. Benchmarking scClassify against alternative methods on cell type classification of the pancreas and PBMC data collection (see also Fig EV2).**

A Performance evaluation for 16 methods on 30 training and test data pairs from the pancreas data collection. Each boxplot ranges from the first to third quartile of classification accuracy for each method with the median of classification accuracy as the horizontal line. The lower and higher whiskers of boxplot are extended to the first quartile minus 1.5 interquartile range and the third quartile plus 1.5 interquartile, respectively.

B Performance evaluation for 16 methods on 84 training and test data pairs from the PBMC data collection. Each boxplot ranges from the first to third quartile of classification accuracy for each method with the median of classification accuracy as the horizontal line. The lower and higher whiskers of boxplot are extended to the first quartile minus 1.5 interquartile range and the third quartile plus 1.5 interquartile, respectively.

C A 1-by-3 panel of dot plots indicating the rankings of each method in 114 pairs of reference and testing data pairs. The x-axis refers to the different methods, and the y-axis refers to the reference and testing data pairs. The dots are coloured by rank, and the size of the dots indicates the degree of accuracy.

was calculated (Fig 3A, blue line, Fig EV3A). The learning curve we estimated (Fig 3A, red line) through this approach exhibited strong agreement ($r = 0.98$, $0.99$) with the validation results, demonstrating the validity of our sample size estimation method (Figs 3B and EV3B).

We applied the sample size estimation to the PBMC datasets (Ding *et al*, 2020; Data ref: Ding *et al*, 2020) and found that different sequencing protocols are associated with different optimal sample sizes. We also observed that while the different protocols demonstrate very similar performance at the first level of the cell type hierarchy, the difference in performance between the protocols became increasingly apparent at the second level of the hierarchy (Fig 3C). These results highlight the importance to consider biotechnological variation between protocols in experimental design.

To investigate the potential influence of capture efficiency, sequencing depth and degree of cell type separation on sample size requirement, we carried out simulations with *SymSim* (Zhang *et al*, 2019) (see Materials and Methods). We found that the within-population heterogeneity has strong effect on the classification performance. scClassify can achieve above 95% average accuracy rate for

the populations with high heterogeneity, while remaining at about 70% average accuracy rate for the extremely homogeneous population (sigma = 1). We observed that the classification accuracy plateaus when the capture efficiency is equal or greater than alpha = 0.02, regardless of the population heterogeneity (Appendix Fig S4A). For a population that is more homogeneous, a higher capture efficiency rate will be required to achieve the greater accuracy rate that scClassify can achieve, as more reference samples are required since scClassify learns relatively more slowly (Appendix Fig S4B). These results together highlight that within-population heterogeneity (sigma) and capture efficiency have a substantial impact on sample size determination, while sequencing depth has much less impact.

Similar observations were found via down-sampling of UMIs across a range of down-sampling proportion parameters ($p_k = 0.1$, ..., 1). We took random draws from a beta-binomial distribution using parameters estimated from the DECENT's beta-binomial capture model (Ye *et al*, 2019) on the PBMC10k data with two cell type levels. We found that for predictions at the first level of the cell type tree, scClassify achieved over 90% accuracy (20 times

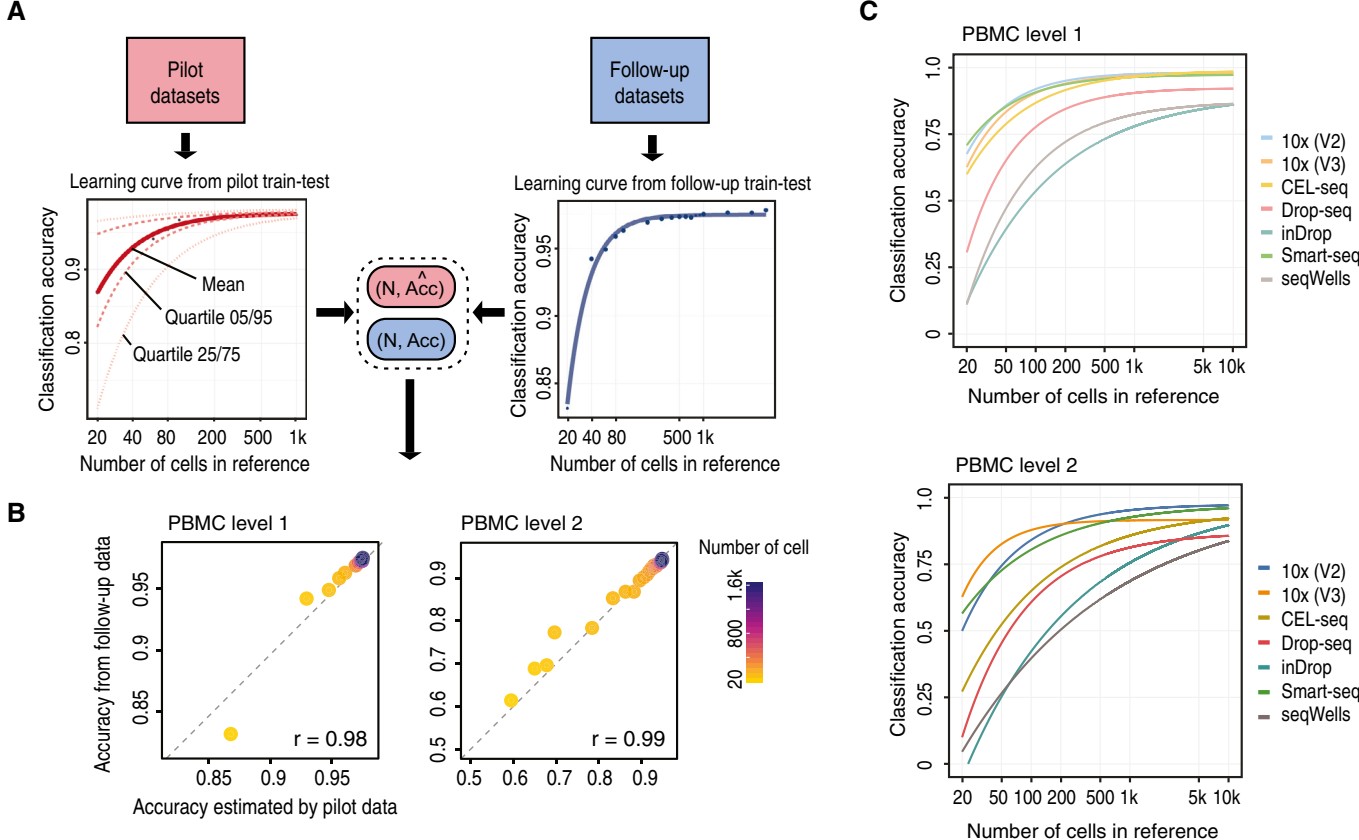

**Figure 3. Sample size estimation for cell type classification (see also Fig EV3).**

A  Schematic illustration of the scClassify sample size learning framework.
B  Scatter plot of sample size estimation based on the pilot data (horizontal axis) compared with accuracy results from the *in silico* experiments (vertical axis).
C  Sample size estimation from the PBMC data collection. Sample size learning curve with the horizontal axis representing sample size (N) and the vertical axis representing classification accuracy. The learning curves for the different datasets provide estimates of the sample size required to identify cell types at the top (top panel) and second (bottom panel) levels of the cell type hierarchical tree.

fivefold cross-validation) with 10% of the original capture efficiency, suggesting discrimination of major (or coarse) cell types is robust to biotechnological variations. In contrast, for prediction at the level 2 of the cell type hierarchy, we found that scClassify requires 50% of the original capture efficiency to achieve a similar level of performance (Fig EV3C) and this has a large impact on sample size estimation. For example, with 20% of the original capture efficiency, scClassify requires a sample size of $N = 80$ and $N = 1,200$ to achieve a 75% classification accuracy at level 1 and level 2 cell type hierarchy, respectively (Appendix Fig S5), highlighting the importance of jointly considering capture efficiency, cell type resolution and intended classification accuracy for constructing a reference dataset.

### *Post hoc* clustering and joint classification further improve cell type annotation

scClassify labels cells from a query dataset as "unassigned" when the corresponding cell type is absent in the reference dataset. With the Xin-Muraro (reference–query) pair (Muraro *et al*, 2016; Data ref: Muraro *et al*, 2016; Xin *et al*, 2016; Data ref: Xin *et al*, 2016),

scClassify correctly identified (Fig 4A) the four shared cell types (alpha, beta, delta and gamma cells) and correctly labelled cells that were only present in the query dataset (i.e. acinar, ductal, stellate, endothelial and delta cells in Muraro dataset) as "unassigned". We then applied scClust (Kim *et al*, 2019) to the "unassigned" cells (Figs 4A and EV4A) and obtained clustering and labelling results consistent with those provided in the Muraro dataset. Another example is the Xin-Wang (reference–query) pair (Wang *et al*, 2016; Data ref: Wang *et al*, 2016; Xin *et al*, 2016), where we found that scClassify is able to correctly label the ductal and gamma cells as "unassigned". After performing *post hoc* clustering and annotation of the clusters using known markers (see Materials and Methods), we found that the final annotated labels were highly consistent with those of the original study (Fig EV4B and C).

Another novel strategy scClassify employs to reduce "unassigned" cells from a query dataset is to include more reference datasets in order to increase sample size in model training while also capturing more or all cell types in the query dataset. Specifically, scClassify utilises multiple reference datasets to perform joint classification where the final prediction is made by weighing across

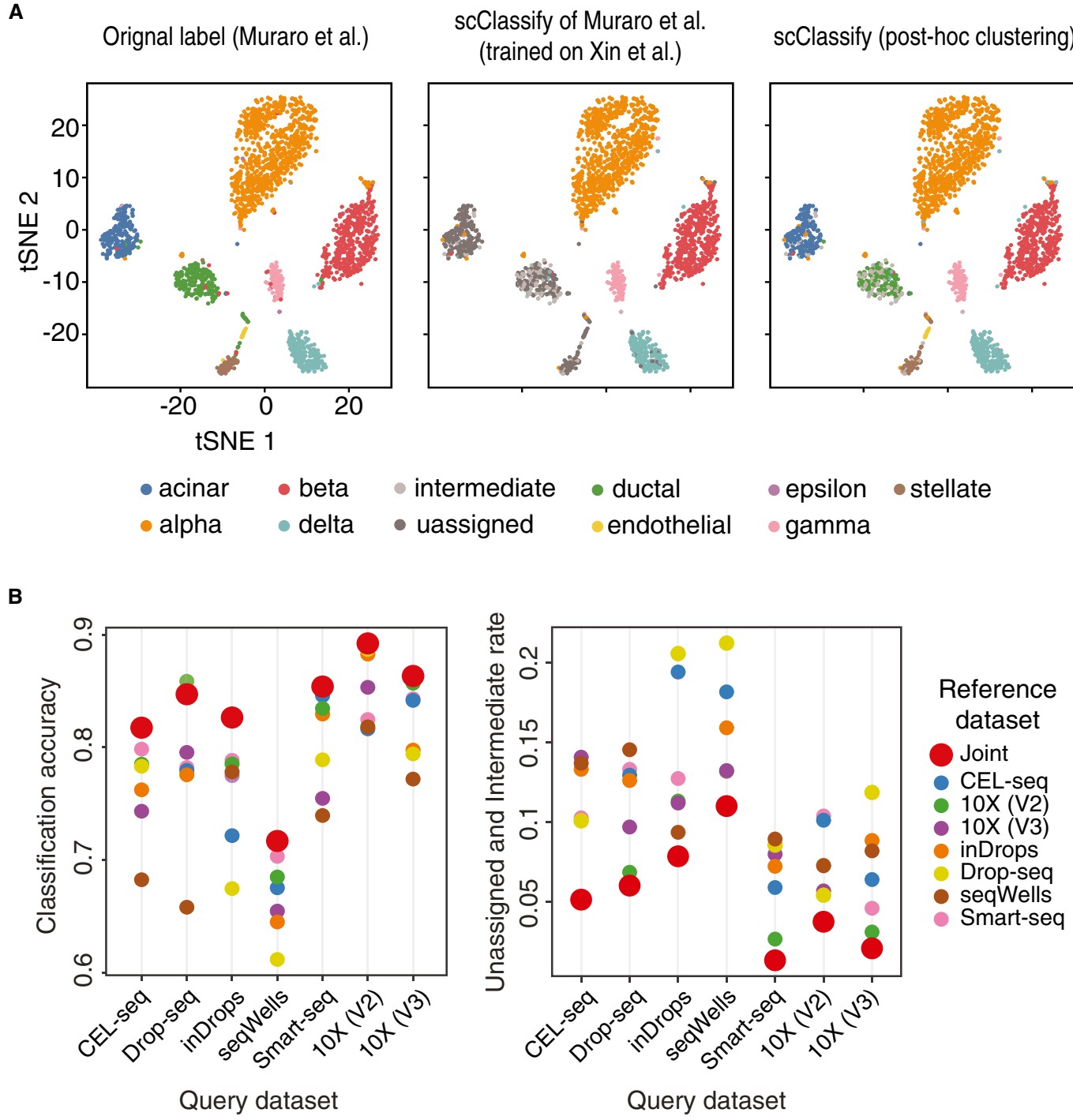

**Figure 4. *Post hoc* clustering of unassigned cells and joint classification of cell types using multiple reference datasets. (see also Fig EV4).**

A   Left panel shows cell types based on the original publication by Muraro *et al* (2016), Data ref: Muraro *et al* (2016). Middle panel shows the predicted cell types from scClassify trained on the reference dataset by Xin *et al* (2016), Data ref: Xin *et al* (2016). Note that the reference dataset does not contain the cell types acinar, ductal and stellate cells. Right panel shows *post hoc* clustering and cell typing results for cells that remained unassigned in the scClassify prediction.

B   Joint classification on the PBMC data collection. Classifying query datasets using the joint prediction from multiple reference datasets (red circle). Classification accuracy as well as unassigned and intermediate rate of the joint prediction is compared to that obtained from using single reference datasets (other colours).

classifiers trained on each reference dataset (see Materials and Methods). The PBMC data collection contains seven datasets each generated using a different sequencing protocol. We applied joint classification on the PBMC data collection by using a "leave-one-protocol-out" approach where datasets generated from all other protocols were used for training and prediction. Compared to the pairwise classification results where a dataset from one sequencing protocol was used for the training and classification of a dataset from a different sequencing protocol, we found that the joint classification results where multiple reference datasets were used resulted in not only higher classification accuracy but also fewer unassigned and intermediate cells compared to when a single reference dataset was used (Fig 4B).

### Scalability of scClassify for large datasets and refinement of cell type annotation through scClassify

To test the scalability of scClassify on large reference and query datasets with complex cell type hierarchies, we used the Tabula Muris FACS dataset (The Tabula Muris Consortium, 2018) as the reference (Appendix Fig S6) and the Microfluidic dataset as the query dataset, collectively amounting to 96,404 cells (Fig 5A). Despite the scale and complexity of the data, we found that scClassify achieved a high classification accuracy of ~ 85%, demonstrating that scClassify scales well to large scRNA-seq datasets. An assessment of performance by tissue type revealed an accuracy of > 80% for most tissue types (nine out of 12 tissues; Fig 5B). Notably, the classification accuracy was 94% for the tongue, 93% for the liver, 92% for the thymus and 92% for the spleen. Next, we found that more than half of the cell types (20 cell types) had a classification accuracy of > 80%, with basal cells, B cells and mesenchymal cells achieving the highest accuracies (Fig 5C).

We further demonstrate the scalability of scClassify on three mouse neuronal datasets, each with about 20 cell types from the visual cortex, generated on three different platforms, and together with over 63,531 cells (Tasic *et al*, 2016; Data ref: Tasic *et al*, 2016; Hrvatin *et al*, 2018; Data ref: Hrvatin *et al*, 2018; Tasic *et al*, 2018; Data ref: Tasic *et al*, 2018) (Appendix Fig S7A–C). We found that cells with common cell type labels in the training and query datasets were predominately correctly annotated by scClassify, with an accuracy in all cases > 90% and an average of 95%. In some cases where the cell type labels did not match between datasets, we found that the annotations derived by scClassify were very closely related to the original labels (Appendix Fig S8A–C).

Finally, we illustrate the ability of scClassify to provide a more refined annotation of cell types in the Tabula Muris Lung Atlas by using as reference dataset the more comprehensive dataset of mouse lung development with 20,931 cells and deeper cell type annotation (22 cell types) (Cohen *et al*, 2018; Data ref: Cohen *et al*, 2018). scClassify showed that stromal cell in the original atlas can be subclassified into smooth muscle and matrix fibroblast cells. To validate our results, we used key cell type markers reported by Cohen *et al* (2018). Specifically, we defined fibroblasts as being marked by a high expression level of Col1a2, smooth muscle cells by Aspn17 and matrix fibroblasts by Macf2 and Mfap4 (Fig 5D). Furthermore, we show that scClassify is able to meet the computational challenge of identifying the presence of small subpopulations of cells in an scRNA-seq dataset. For

example, scClassify was also able to annotate three minor populations: pericytes (19 cells, originally labelled as stromal cells), ciliated cells (21 cells) and neutrophils (28 cells, originally labelled as leucocytes). The new classifications were supported by the high expression level of marker genes for each cell type (pericytes: Gucy1a3; neutrophils: Retnlg; Fig 5D). Remarkably, scClassify was able to identify a group of only six cells that were originally labelled as leucocytes and dendritic cells but express high levels of the three top basophil markers, Ccl3, Ccl4 and Ifitm1 (Fig 5D) (Cohen *et al*, 2018). This illustrates the ability of scClassify to identify cell types present in very small numbers in the data, something which is usually extremely challenging to do with clustering alone.

## Discussion

In supervised cell type classification, the performance of methods may be influenced by various characteristics in the data. For example, better quality reference and query datasets potentially lead to better performance in classification. Furthermore, the more heterogeneous the cell types are in the reference data, the easier they can be distinguished by the classification algorithm. This can be seen in our sample size simulation study, where we found that the within-population heterogeneity strongly affects classification performance (Appendix Fig S4).

The choice of evaluation metric is important for comparing performance of supervised cell type classification methods. In this study, we have extended an evaluation framework proposed by de Kanter *et al* (2019) to divide the prediction results into seven different categories. Compared to the traditional binary classification evaluation metrics such as the F1 score used in other benchmark studies (Abdelaal *et al*, 2019; Zhao *et al*, 2019), our evaluation framework aims to capture the complexity in our cell type classification results. For instance, categories of "incorrectly assigned" and "correctly unassigned" will evaluate whether a method is able to accurately classify a cell whose cell type is not present in the reference data, a common scenario that cannot be properly quantified using a metric such as F1 score. Capturing various complex scenarios in single-cell classification in the evaluation metric could provide a better understanding of the method performance such as these visualised in Fig EV2A.

To summarise, scClassify is a robust supervised classification framework for comprehensive and accurate cell type annotation. scClassify addresses several limitations of existing supervised classification methods by generating multilevel cell type annotations through cell type hierarchies constructed from single or multiple reference datasets. While the cell type hierarchies enable the interpretation of cell types at multiscale, the joint classification of cells using multiple references allows the representation of a broad range of cell types that would be unachievable through a single dataset, enabling classification of cells that would otherwise be misclassified or unassigned. Importantly, in cases where the sample size of a given cell type in the reference dataset is too small, scClassify allows the query cell of the corresponding cell type to be annotated as an intermediate cell type and therefore does not force the classification of cells when it cannot be

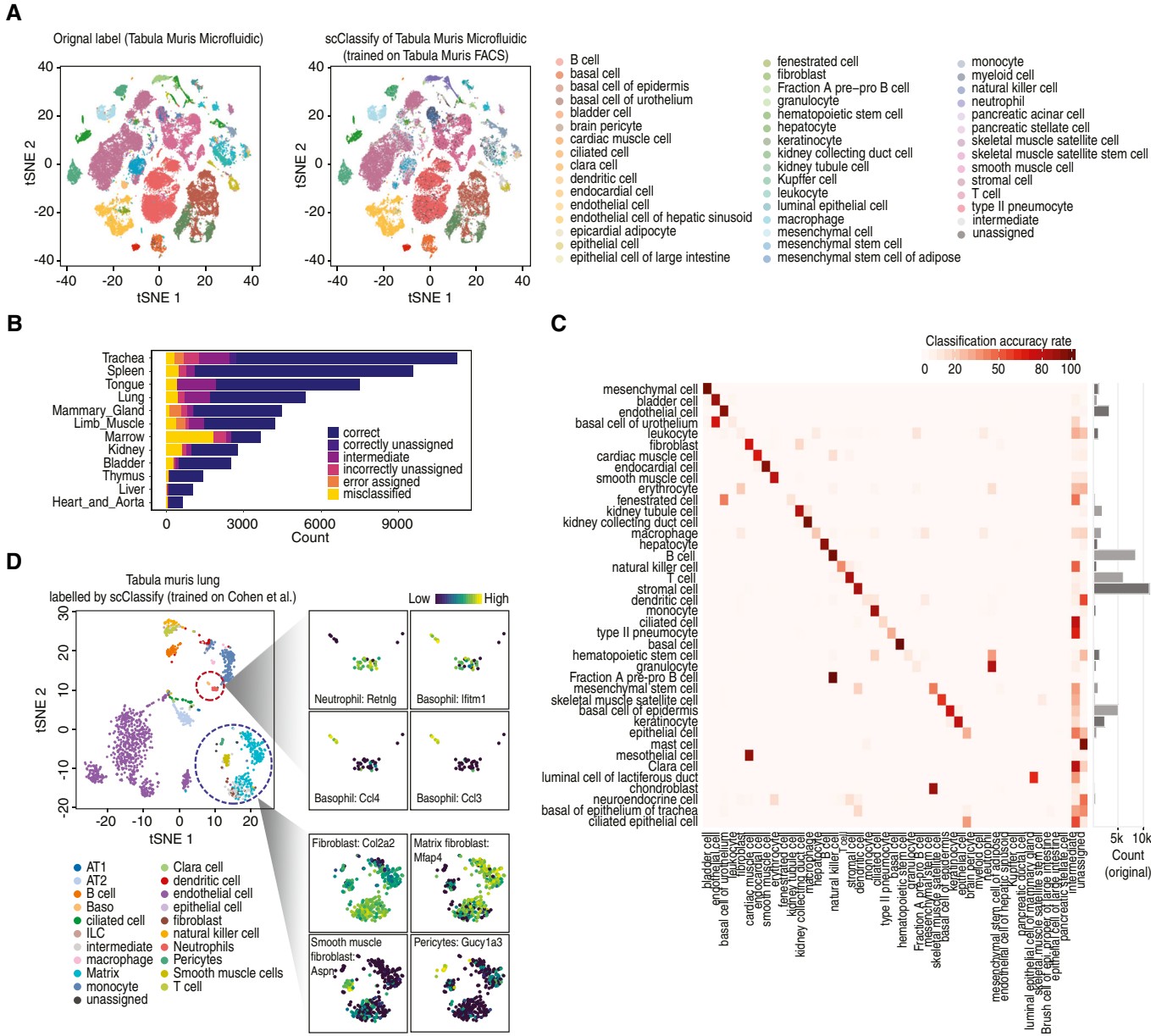

**Figure 5. Cross-platform classification of cell types using scClassify.**

A   tSNE visualisation of the Tabula Muris Microfluidic dataset (The Tabula Muris Consortium, 2018; Data ref: The Tabula Muris Consortium 2018). Cell typing is based on either the original publication (left panel) or scClassify prediction (right panel). scClassify was applied to the Tabula Muris Microfluidic data collection with the Tabula Muris FACS dataset as reference data for model training.

B   Bar plot indicating the predicted cell types organised by tissue types when the Tabula Muris Microfluidic dataset was used as query and the Tabula Muris FACS dataset was used as reference.

C   Heatmap of data in (A) comparing the original cell types given in the Tabula Muris Microfluidic data (rows) against the scClassify predicted cell types (columns) generated using the Tabula Muris FACS data as the reference dataset.

D   scClassify prediction results of cells in lung tissue type in the Tabula Muris FACS data by using Cohen et al dataset as reference (Cohen et al, 2018; Data ref: Cohen et al, 2018). The large tSNE plot is the full data coloured by scClassify predicted cell types, while the smaller tSNE panels show two subsets of cells from the large plot, each coloured to highlight four marker genes, where the lighter yellow colour represents higher gene expression.

confidently assigned to a terminal cell type. Rather, it implements a *post hoc* clustering procedure for discovery of novel cell types among the unassigned cells. Collectively, the capacity of scClassify to perform multiscale cell type classification and joint classification from multiple reference datasets, to estimate sample size and to implement *post hoc* clustering enables accurate and nuanced identification of cell types and their associated genes from scRNA-seq data.

# Materials and Methods

## Reagents and Tools table

| Reagent/Resource | Reference or source | Identifier or Catalog Number |
|---|---|---|
| **Software** | | |
| R version 3.6 | https://www.r-project.org/ | N/A |
| Python version 3.7.3 | https://www.python.org/ | N/A |
| ACTINN GitHub version c3dd085 | https://github.com/mafeiyang/ACTINN | N/A |
| CHETAH version 1.1.2 | https://github.com/jdekanter/CHETAH | N/A |
| CaSTLe GitHub version 258b278 | https://github.com/yuvallb/CaSTLe | N/A |
| Garnett version 0.1.4 | https://cole-trapnell-lab.github.io/garnett/ | N/A |
| SingleR version 1.0.1 | https://github.com/dviraran/SingleR | N/A |
| Moana version 0.1.1 | https://github.com/yanailab/moana | N/A |
| scID version 0.0.0.9000 | https://github.com/BatadaLab/scID | N/A |
| scPred version 0.0.0.9000 | https://github.com/IMB-Computational-Genomics-Lab/scPred | N/A |
| scVI version 0.3.0 | https://github.com/YosefLab/scVI | N/A |
| scmap version 1.1.6 | https://bioconductor.org/packages/release/bioc/html/scmap.html | N/A |
| SingleCellNet version 0.1.0 | https://github.com/pcahan1/singleCellNet/ | N/A |
| SVMreject version 0.22.2 | https://scikit-learn.org/stable/modules/generated/sklearn.svm.LinearSVC.html | N/A |
| scater version 1.13.7 | https://bioconductor.org/packages/release/bioc/html/scater.html | N/A |
| limma version 3.42.0 | https://bioconductor.org/packages/release/bioc/html/limma.html | N/A |
| mixtools version 1.1.0 | https://cran.r-project.org/web/packages/mixtools/index.html | N/A |
| minpack.lm version 1.2-1 | https://cran.r-project.org/web/packages/minpack.lm/index.html | N/A |
| Bench 1.1.1 | https://cran.r-project.org/web/packages/bench/index.html | N/A |
| utils 3.6.2 | https://www.rdocumentation.org/packages/utils/versions/3.6.2/topics/Rprofmem | N/A |
| Memory-profiler 0.57.0 | https://pypi.org/project/memory-profiler/ | N/A |

## Methods and Protocols

### scClassify framework

scClassify is a classification framework for identifying cell types in scRNA-seq datasets. It uses a _cell type hierarchy_ constructed from a reference dataset and ensemble learning models at each _branch node_ of the hierarchy trained on the reference dataset (Fig 1A). Specifically, the hierarchical ordered partitioning and collapsing hybrid (HOPACH) algorithm (van der Laan & Pollard, 2003) is used to construct the hierarchical cell type tree from the reference dataset (see Cell type tree section). In contrast to standard hierarchical clustering, this method allows a parent node to be partitioned into multiple child nodes, which is more consistent with the natural progression from broad to more specific cell types, where a cell type can have two or more subtypes.

We will use the term _training set_ interchangeably with _reference set_ (or _sets_) and restrict the usage of the term _test set_ to situations in which the true cell types of the cells whose annotation is being predicted are known, i.e. to situations where we are assessing or comparing the performance of scClassify. We will use the term _query set_ or _query cell_ when the unknown types of cells are to be predicted, and we write _test/query_ when we wish to cover both cases.

scClassify uses a weighted _k_NN classifier. More weight is given to neighbouring cells that are nearer, as defined by a similarity metric, to a query cell. To incorporate a variety of information, six similarity metrics and five cell type-specific gene selection methods are used in scClassify (see Ensemble of base classifiers section). Each of the 30 base classifiers is trained using one of the similarity metrics and one of the gene selection methods. The final predictions are made by an ensemble classifier that weights individual classifiers depending on their training error. The best base classifier will have the highest weight, while a classifier with < 50% accuracy will have negative weight.

An ensemble classifier is trained at each branch node of the cell type tree constructed using HOPACH, where both the training and test/query phase follow a top-down approach. That is, for parent nodes, classifiers are trained using the cells belonging to their child nodes. To characterise nodes at different levels, we carry out the gene selection step separately at each level of the tree. In the test/query step, a cell will have its type predicted from the highest to the lowest level, as long as it is possible to make a satisfactory prediction at every level (see Multilevel classification section). If that is not possible, the cell will be given an intermediate type. Finally, we carry out _post hoc_ clustering of unassigned cells.

### Component 1: Cell type tree

To construct the cell type tree from a reference dataset, we first take the union across cell types of all the sets of genes found using *limma* (Ritchie *et al*, 2015) to be differentially expressed (DE) between one cell type and all other cell types (one vs all). We next use HOPACH on the average expression of selected genes from each cell type to construct the cell type tree. Starting from the tree *root*, the maximum number of children at each *branch node* was set as 5 by default and can be modified when data containing a large number of cell types are used as reference. The root of the tree consists cells of all the cell types; the branch nodes of the tree are the less refined cell types, as each may contain multiple subtypes; at the bottom level of the tree are the *leaves*, representing the finest cell types in the reference dataset. Note that since we build the cell type tree using the average expression of the cell type, the tree built by the HOPACH algorithm is robust in general.

### Component 2: Ensemble of base classifiers

At each branch node, an ensemble classifier is built from 30 base classifiers. Each is a weighted *k*NN model with a different combination of similarity metric and gene selection methods (Fig 1A). These 30 base classifiers are all combinations of six similarity metrics and five cell type-specific gene selection methods. The motivation is that while numerous similarity metrics are available, they often lead to different results, reflecting the different properties of the data each metric is measuring (Kim *et al*, 2019; Skinnider *et al*, 2019). We use Pearson's correlation, Spearman's correlation, Kendall's rank correlation, cosine distance, Jaccard distance and weighted rank correlation (Iman & Conover, 1987). Since not all genes are cell type-specific (Lin *et al*, 2019), only the ones that are informative for cell type classification should be included in a similarity metric. There are various computational methods for selecting genes that are cell type-specific (Strbenac *et al*, 2016). Five of them are included in scClassify for the different types of information each of them provides, four being based on two-sample tests. These are DE genes using *limma* (Ritchie *et al*, 2015), differentially variable (DV) genes using Bartlett's test, differentially distributed (DD) genes using the Kolmogorov–Smirnov test, bimodally distributed (BD) genes using the bimodality index (Wang *et al*, 2009) and genes with DE proportions (DP) using a chi-squared test. To make a consensus prediction from an ensemble, we weight each base classifier in a manner similar to that in AdaBoost (Bauer & Kohavi, 1999). Specifically, the weight of each classifier is calculated as follows:

$$\alpha_t = \ln \frac{1 - \epsilon_t}{\epsilon_t},$$

where $\epsilon_t$ is the error rate achieved by training and testing the base classifier $t$ on the reference dataset, $t = 1, \ldots, 30$. Let $l_{jt} \in \mathcal{L}$ be the cell type label predicted for cell $j$ by base classifier $t$, where $\mathcal{L}$ is the set of all the possible labels available to scClassify, including intermediate node labels and "unassigned". The ensemble prediction for cell $j$ is made as Bauer and Kohavi (1999):

$$l_j^* = \arg\max_{c \in \mathcal{L}} \sum_{t:l_{jt}=c} \alpha_t.$$

Note that the weight for each base classifier can also be flexibly specified by the user or based on a pre-defined classification accuracy threshold in the scClassify function for up- and down-weighing or pruning base classifiers.

### Weighted kNN

To relate the cell type predicted for a test/query cell to that of its nearest neighbours in the reference dataset, we use a distance-weighted *k*NN classifier (Dudani, 1976). Let $T = \{(x_j, c_j)\}_{j=1}^N$ denote the reference data for a branch node, where $x_j \in \mathcal{R}^m$ is the expression vector of cell type-specific genes for cell $j$, and $c_j$ is the corresponding cell type label. For a test/query cell with expression vector $\mathbf{y} \in \mathcal{R}^m$, we first calculate the distances between it and all cells in the training dataset. Let $d_k$ be the distance between such a cell and its $k$-th nearest neighbour in the reference dataset, $k = 1,\ldots, N$ (See Appendix Table S2). We then identify the $K$-nearest cells in the training dataset for this cell (By default, $K = 10$). A weight $w_j$ attributed to the $k$-th nearest neighbour is defined as

$$w_k = \begin{cases} \frac{d_K - d_k}{d_K - d_1}, & d_K \neq d_1 \\ 1, & d_K = d_1 \end{cases}.$$

The query cell is then predicted to have the cell type with the greatest total weight, i.e. we use weighted majority voting.

### Discriminative genes

To characterise branch nodes in the tree, we select genes at each node separately by comparing the gene expression levels in each cell type at a node against those of all other cell types at that node using the following five metrics:

- Differentially expressed genes: Differential expression analysis is carried out using limma-trend (Ritchie *et al*, 2015). Using the *topTable* command of the *limma* package, we extract the genes with fold change > 1.
- Differential variable genes: DV analysis is carried out using Bartlett's test.
- Differentially distributed genes: DD analysis is carried out using the Kolmogorov–Smirnov test.
- Differentially proportioned genes: DP analysis is performed by a chi-squared test. Here, each gene is classified as expressed or not expressed in each relevant cell, where expressed means it has an expression level greater than a certain threshold (by default, this threshold is set at 1 for log-transformed data). The difference between the proportion of cells in one cell type in which a gene is expressed and that proportion across all other cell types at that node is denoted as "proportion difference".
- Bimodally distributed genes: We calculated the bimodality index for each gene (Wang *et al*, 2009), as follows

$$BI = \frac{|m_1 - m_2|}{s\sqrt{p(1-p)}},$$

where $m_1$ and $m_2$ are the means of the expression levels of one cell type and of all the other cell types, $s$ is the (assumed) common standard deviation, and $p$ is the proportion of cells of the cell type under consideration. The genes are then ranked based on the bimodality index.

For each method, genes are first ranked according to their adjusted *P*-values, and then, a maximum of 50 top-ranked genes whose adjusted *P*-values are < 0.01 and the proportion difference (defined in 4) > 0.05 are selected from each method. These genes are included in training model.

### Component 3: Multilevel classification

Starting from the root of the cell type tree, scClassify calculates the distances between a query cell and all reference cells at and below a branch node and classifies the query cell to a child of that node if the following two criteria are fulfilled. First, the nearest neighbour cells must have correlations higher than a certain threshold. This is determined using a mixture model on the correlation of the cell type of this nearest neighbour, using the *normalmixEM* function in the package *mixtools* (Benaglia *et al*, 2009). Second, the weights of its assigned cell type must be larger than a certain threshold, whose default is set at 0.7. Cells that fail to pass either of these criteria will not be classified to the next level. Cells that do not progress above the root will be considered *unassigned*. Those cells that are classified at a branch node, but whose classification process does not reach a leaf of the tree, will be viewed as having an *intermediate* cell type. In such cases, the final type of a query cell is defined by the last assigned branch node, where a branch node cell type is defined as the collection of cell types of all its child nodes. Finally, cells that reach the leaf level will be labelled by the cell type each leaf represents.

### Component 4: Post hoc clustering of unassigned cells

scClassify uses a modified version of the SIMLR algorithm to cluster unassigned cells (Wang *et al*, 2017; Kim *et al*, 2019). To illustrate scClassify's capacity to annotate cells that are not in the reference dataset, we trained scClassify on a reference dataset that had only four cell types (Xin *et al*), and used this to predict cell types for a dataset with nine cell types from human pancreas (Muraro *et al*). Cells that were classified as unassigned were then clustered and further identified in the familiar way following clustering. Specifically, DE genes in each cluster (one vs all) were found using *limma* (Ritchie *et al*, 2015). Each cluster was subsequently annotated based on its DE genes and the markers provided by Muraro *et al* in the paper (i.e. acinar: PRSS1; ductal: SPP1 and KRT19; stellate: COL1A1 and COL1A2; endothelial: ESAM; delta: SST; Fig EV4A). Similarly, we performed *post hoc* clustering on the human pancreas dataset generated by Wang *et al*, where most of ductal and stellate cells are correctly called unassigned by scClassify (Fig EV4B). We annotated each cluster based on its genes DE when compared to other clusters (ductal: SPP1; stellate: COL1A1 and COL1A2; Fig EV4C). In both cases, we found that the cell types assigned by scClassify were highly consistent with those originally published. Note that other clustering methods are also applicable in the *post hoc* clustering procedure.

### Component 5: Joint classification using different training datasets

When multiple scRNA-seq datasets are available, each profiling the same tissues or including overlapping cell types, scClassify makes use of them all by training a collection of ensemble models using each dataset and makes predictions for a query dataset using a joint classification method where predictions from each reference dataset are weighted by their training error. This is because classifiers with

larger training errors usually make poorer predictions, and therefore should be down-weighted. For each query cell, scClassify assigns the cell type with the largest average score using the ensemble models from each of the reference datasets.

### Data collections and processing

The datasets used in this study are publicly available and include the following:

- The pancreas data collection was downloaded from the National Center for Biotechnology Information (NCBI) Gene Expression Omnibus (GEO) for GSE81608 (Xin *et al*, 2016; Data ref: Xin *et al*, 2016), GSE83139 (Wang *et al*, 2016; Data ref: Wang *et al*, 2016), GSE86469 (Lawlor *et al*, 2017; Data ref: Lawlor *et al*, 2017), GSE85241 (Muraro *et al*, 2016; Data ref: Muraro *et al*, 2016), GSE84133 (Baron *et al*, 2016; Data ref: Baron *et al*, 2016) and EBI ArrayExpress website for E-MTAB-5061 (Segerstolpe *et al*, 2016; Data ref: Segerstolpe *et al*, 2016).
- The PBMC data collection was downloaded from the Single Cell Portal with accession numbers SCP424 (Ding *et al*, 2020; Data ref: Ding *et al*, 2020), which contains a collection of seven datasets that were sequenced using different platforms (Smart-Seq, CEL-Seq, inDrops, dropSeqs, seqWells, 10x Genomics (V3) and 10x Genomics (V2)).
- The Tabula Muris mouse data were downloaded from https://tabula-muris.ds.czbiohub.org/ (The Tabula Muris Consortium, 2018; Data ref: The Tabula Muris Consortium, 2018).
- The neuronal data collection was downloaded from GEO accession number GSE71585 (Tasic *et al*, 2016; Data ref: Tasic *et al*, 2016), GSE115746 (Tasic *et al*, 2018; Data ref: Tasic *et al*, 2018) and GSE102827 (Hrvatin *et al*, 2018; Data ref: Hrvatin *et al*, 2018).
- The mouse lung development dataset was downloaded from GEO accession number GSE119228 (Cohen *et al*, 2018; Data ref: Cohen *et al*, 2018).
- The PBMC10k data collection generated by Cell Ranger version 3.0.0 was downloaded from 10x Genomics website: https://support.10xgenomics.com/single-cell-gene-expression/datasets/3.0.0/pbmc_10k_v3.

We organised these datasets into large data collections for either performance assessment or use as case studies (Fig EV1B). For the six pancreas datasets (Baron *et al*, 2016; Muraro *et al*, 2016; Segerstolpe *et al*, 2016; Wang *et al*, 2016; Xin *et al*, 2016; Lawlor *et al*, 2017), we manually checked the cell type annotations that were provided by the original authors of each dataset and curated the labels such that the naming convention is consistent across datasets. For example, the cell type "PP" in Xin dataset was changed to "gamma", as "gamma" was the name used by all other datasets. Similarly, "mesenchymal" in Muraro and Wang was changed to "stellate". We also removed the cell types that were labelled as "co-expression" and "unclassified endocrine" in Segerstolpe dataset, while for Baron, we grouped quiescent stellate and activated stellate as stellate.

For all datasets described in Fig EV1B, only cells that passed the quality control of the original publication and assigned cell types were included. We performed size factor standardisation to the raw count matrices for each batch/dataset using the *normalize* function in the R package *scater* (McCarthy *et al*, 2017) and used the

log-transformed gene expression matrices as inputs to scClassify and all other methods. For the PBMC10K data, we removed the doublets from the dataset using *DoubletFinder* (McGinnis *et al*, 2019). We labelled the cells following the approach proposed by 10x Genomics for PBMCs (Zheng *et al*, 2017), which allowed us to annotate 11 PBMC cell types. In sample size calculation study, we considered "B", "Monocyte", "T", "NK" and "CD34$^+$" cell type as first-level coarse annotation and then expanded "T" to "CD4$^+$ T" and "CD8$^+$ T" for second-level finer annotation.

### Performance evaluation

We extend the evaluation framework introduced in CHETAH (de Kanter *et al*, 2019) by describing the results of our predictions as "correctly classified", "misclassified", "intermediate" (correct and incorrect), "incorrectly unassigned", "incorrectly assigned" and "correctly unassigned" (Fig EV1A). For cells of a given cell type from a query dataset, we first consider whether this cell type is present in the reference dataset. If it is, we then consider whether these cells are classified to the leaf level in the cell type hierarchy. Cells that are classified to the leaf level are "correctly classified" if their predicted cell types match their annotated cell types in the original study; otherwise, they are "misclassified". Of cells not classified to the leaf level, we consider the unassigned to be "incorrectly unassigned", while if they are assigned "intermediate" cell types, we check whether their "intermediate" types contain the annotated cell types from the original study. Those that are on the correct branch of the cell type tree are "correct intermediate"; otherwise, they are considered to be "incorrect intermediate". Cells from a cell type that in the query dataset that is not present in the reference dataset can be either "incorrectly assigned" to a cell type in the reference dataset or "correctly unassigned".

### Benchmarking and method comparison

To evaluate and compare the performance of scClassify, we obtained 14 other publicly available scRNA-seq classification methods (Appendix Table S1). These packages were installed either through their official CRAN or Bioconductor website, where available, or from their GitHub page. Two collections of datasets, pancreas and PBMC, were used (Fig EV1B).

For the pancreas datasets, we defined the *hard* cases to be the 14 (training, test) = (reference, query) set pairs, where the reference/training dataset has *fewer* cell types than the query/test dataset. The remaining 16 pairs were called *easy*. The evaluation of scClassify on the PBMC datasets was carried out at two levels of the cell type hierarchy, coarse ("level 1") or fine ("level 2"). Each led to 42 (training, test) set pairs. For level 1, we combined CD16$^+$ and CD14$^+$ monocytes into the cell type of monocytes, and we combined CD8$^+$ and CD4$^+$ T cells into the cell type of T cells. For the level 2, we used the original cell types.

The evaluation procedure was as follows. For both the pancreas and the PBMC data collections, we used one of the datasets as reference or training set, and all the other datasets as query or test sets on which the accuracy of cell type predictions could be calculated. This gives us $6 \times 5 = 30$ distinct (training, test) set pairs of pancreas datasets and $7 \times 6 = 42$ distinct pairs of PBMC datasets for each of the two levels.

For each of the 14 methods, we evaluated the performance of a total of 114 (training, test) pairs. We used the default settings given

in the package README or vignette for training each method. The Garnett method requires the specification of marker genes for model training. In this case, we benchmarked using two approaches. In the first approach, marker genes were obtained from the published marker file from Garnett website and from published literature (Pang *et al*, 2005; Scarlett *et al*, 2011; Muraro *et al*, 2016; Lawlor *et al*, 2017; Collin & Bigley, 2018). In the second approach, the list of DE genes was obtained from scClassify by training on the same dataset and then used as the input marker genes. Note that for benchmarking, we did not include the time and memory usage to generate the DE genes from scClassify. Another method, SVMreject, requires a pre-specified threshold for classifying samples into the unassigned category. In this case, we used the same threshold of 0.7 as in a previous published benchmarking paper (Abdelaal *et al*, 2019). Cells with predicted probability lower than this threshold are determined to be unassigned. The log-transformed size factor-normalised gene expression matrix was used as the input for all models.

The results were the types predicted for all cells in the test/query datasets. To calculate classification accuracy, we compared the predicted cell types with those provided in the reference dataset. Methods that do not allow unassigned or intermediate prediction were assessed based on cells that are "correctly classified" and "misclassified" (Fig EV1A). For methods that do allow unassigned predictions, "incorrectly unassigned", "incorrectly assigned" and "correctly unassigned" were also included for calculating classification accuracy. Finally, for methods that allow both unassigned and intermediate predictions, we considered "correct intermediate" as correct predictions in calculating classification accuracy.

### Memory and running time comparison
#### Dataset preparation
We utilised the Tabula Muris dataset to benchmark the memory and running time by combining the Microfluidic and FACS datasets via the commonly expressed genes (The Tabula Muris Consortium, 2018). We chose the 16 cell types with the greatest numbers of cells for subsequent processing. The dataset was then used to generate two benchmarking datasets.

1  To benchmark training and test time, we randomly sampled cells and created datasets with 100, 200, 500, 1,000, 2,000, 5,000, 10,000, 20,000 and 30,000 cells.
2  To benchmark the impact of the number of cell types, we created datasets with 4, 6, 8, 10 and 12 cell types, while keeping the total number of cells in each dataset the same at 5,000 cells.

#### Memory and running time evaluation
All evaluation was performed on a research server with dual Intel (R) Xeon(R) Gold 6148 Processor (40 total cores, 768 GB total memory) and dual RTX2080TI GPUs. The following dataset set-up was utilised for the evaluation:

- To evaluate the impact of the number of cells in the training sample, we set the training dataset to be 100, 200, 500, 1,000, 2,000, 5,000, 10,000, 20,000 and 30,000 cells and kept the test dataset the same at 2,000 cells.
- To evaluate the impact of the number of cells in the testing sample, we performed the reverse, where the test dataset had 100,

200, 500, 1,000, 2,000, 5,000, 10,000, 20,000 and 30,000 cells and the training dataset was kept at 2,000 cells.

- To evaluate the impact of the number of cell types in the training dataset, we kept the number of cell types in the training dataset as 4, 6, 8, 10 and 12 and the number of cells in both the training and test datasets was kept at 5,000 cells.

For benchmarking R-based methods, we measured the total allocated memory and system running time using the *Bench* package. The SingleR method has a built-in parallel framework and cannot be measured using *Bench*. In this case, we used the function *Rprofmem* in *utils* package and measured system running time using the R built-in *Sys.time* function (R Core Team, 2019).

For benchmarking Python-based methods, we measured the total allocated memory using the *tracemalloc* function, which is a Python built-in function. The system time was measured using the Python built-in *time* function. The method ACTINN executes its function in bash terminal and cannot be measured using *tracemalloc*. For this method, we used the package *Memory Profiler*. The system running time was still measured using the *time* function.

### Framework for sample size estimation and sensitivity investigation of reference data

#### Learning curve construction

Estimating the number of cells required in a scRNA-seq study in order to discriminate between two given cell types for a given platform is essentially sample size determination for a high-dimensional classification problem. Our approach is to construct a learning curve (Cortes *et al*, 1994; Mukherjee *et al*, 2003) based on a pilot study and estimate the empirical classification accuracy as a function of the size of the reference dataset. For each dataset of size $N$, we randomly divide the data into a training set of size $n$ and a test set of size $N - n$, independently $T$ times ($T = 20$), with the resulting accuracy rates being denoted by $\{a_{n,t}\}_{t=1}^{T}$. The average accuracy rate was calculated by $a_n = 1/T \sum_t a_{n,t}$. We then fit an inverse power law function of sample size to the average accuracy rate as follows:

$$a_n = bn^{-c} + \alpha,$$

where $b$ is the learning rate, $c$ is the decay rate, and $\alpha$ is the maximum accuracy rate that can be achieved by the classifier.

When we did this, we noticed that the inverse power law function gave a poor fit in some cases where we had a small sample size. Therefore, we moved to a two-component inverse power law function to relate average accuracy rate to sample size, separately fitting to small and large sample sizes. That is, we now fit to

$$a_n = 1(n<d)b_1 n^{-c_1} + 1(n \geq d)b_2 n^{-c_2} + \alpha,$$

where $d$ here is the transition point of the model, $b_1$ and $b_2$ are the learning rates of each component, and $c_1$ and $c_2$ are their decay rates.

The function *nlsLM* in R package *minpack.lm* is used to fit both learning curves models (Elzhov *et al*, 2016). To fit the mixture model, we first fixed $d$ and estimated $b_1$, $b_2$, $c_1$, $c_2$ and $\alpha$. After fitting the model for various values of $d$, we determined the best model to be the one with the smallest residual sum of squares.

#### Evaluation of the learning curve

To validate the estimated learning curve, we randomly divide the PBMC10k data into the pilot (20%) and validation (80%) sets. The pilot set represents data we might have from a typical pilot study on which sample size estimation is based. The validation set represents data generated from an actual study. For this evaluation, the validation set is further divided into a training set and an independent test set, to calculate the accuracy of our learning curve for any given sample size. We hope and expect that the learning curves constructed from these two subsets of the data are consistent, and we assess their consistency by the Pearson correlation between the accuracies estimated from the pilot data and those estimated from the validation data.

### Sensitivity investigation of capture efficiency and sequencing depth via simulation

To investigate the potential influence of capture efficiency, sequencing depth and degree of cell type separation on the sample size requirement, we carried out simulations using *SymSim* (Zhang *et al*, 2019), estimating the parameters on PBMC10k data, with following parameter settings:

- Within-population heterogeneity ($\sigma$) from 0.2 to 1 in increments of 0.2. According to the definition by *SymSim*, a higher $\sigma$ indicates a more homogeneous simulated population.
- Capture efficiency at the following values 0.001, 0.01, 0.02, 0.03, 0.04, 0.05, 0.06, 0.07, 0.08, 0.09 and 0.1.
- Sequencing depth at the following values 30,000, 80,000, 160,000, 300,000 and 500,000.

### Sensitivity investigation of capture efficiency and sequencing depth via down-sampling

We first fitted the DECENT (Ye *et al*, 2019) model on the UMI data matrix for the PBMC10k data collection. After obtaining the parameter estimates of DECENT's beta-binomial capture model, we conducted a UMI down-sampling using random draws from a beta-binomial distribution. Specifically, for the UMI raw count $x_{ij}$ of gene $i$ and cell $j$, the down-sampled matrix $Z^{(k)}$ with down-sampling parameter $p_k$ is generated by

$$z_{ij}^{(k)} \sim \text{Beta-binomial}(x_{ij}, \rho_{ij}, p_k),$$

where $\rho_{ij}$ is the correlation parameter of beta-binomial distribution, calculated using

$$\log\left(\frac{\rho_{ij}}{1 - \rho_{ij}}\right) = \tau_0 + \tau_1 \log(x_{ij} + 0.1),$$

where $\tau_0$ and $\tau_1$ are estimates of DECENT's of capture model parameters. Within each cell, the parameter $p_k$ in this case can be interpreted as the ratio of capture efficiency in the down-sampled dataset relative to the original dataset.

We carried out fivefold cross-validation 20 times with each down-sampling proportion parameter, which ranges from 0.1 to 1 on PBMC10k data with two cell type levels. We found that for predictions at the top of cell type tree, scClassify achieved over 90% accuracy, even with 10% of original capture efficiency. For prediction at the second level of the cell type hierarchy, it requires 50% of

original capture efficiency to achieve a similar level of performance (Appendix Fig S5). We then constructed the learning curves of down-sampling matrices for $p_k$ = 0.2, 0.5, 0.8, 1. As shown in Appendix Fig S6B, for level 1 prediction, learning curves converge to the highest accuracy rate at $N = 80$ when $p_k$ is greater or equal to 0.5. However, for level 2 prediction, it requires $N = 400$, and for the cases with 20% capture efficiency, the accuracy converges to about 75%.

### Sensitivity analysis of scClassify
#### Robustness and stability of cell type classification
To evaluate the robustness and stability of scClassify on cell type classification, we perturbed each training dataset in the pancreas data collection by randomly subsampling 80% of the cells and testing the impact of this on classifying each test dataset (a total of 30 training and test set pairs). This procedure was repeated 10 times for each training and test pair.

#### Number of nearest neighbours considered in weighted kNN
To evaluate the performance of scClassify on using different numbers of nearest neighbours in the weighted $k$NN classifiers (denoted as $k$ in scClassify function), we performed scClassify with various values of $k$, ranging from 5 to 20 on the 30 training and test pairs of pancreas datasets.

#### Maximum number of children per branch node in HOPACH tree
A key hyperparameter in HOPACH algorithm is the choice of the maximum number of children at each node (denoted as *hopach_kmax* in scClassify function). To investigate how different values of *hopach_kmax* affect the performance of scClassify, we performed scClassify on 30 (training, test) pairs of pancreas datasets with *hopach_kmax* = 3, 5, 7, 9, 11.

#### Correlation threshold
By default, the choice of correlation threshold in scClassify is determined dynamically by a mixture model on the distribution of correlations. We compared this dynamic threshold with a range of pre-defined thresholds (0, 0.1, 0.2, 0.3, 0.4, 0.5, 0.6, 0.7 and 0.8) by performing scClassify on the 30 training and test pairs of pancreas datasets.

## Data availability

An open-source implementation of scClassify in R is available from https://github.com/SydneyBioX/scClassify. Code to reproduce all the analyses presented is available at https://github.com/SydneyBioX/scClassify_analysis.

**Expanded View** for this article is available online.

## Acknowledgements

The authors thank all their colleagues, particularly at The University of Sydney, School of Mathematics and Statistics, for their support and intellectual engagement. We also thank Andy Tran for testing the package. The following sources of funding for each author, and for the manuscript preparation, are gratefully acknowledged: Australian Research Council Discovery Project Grant (DP170100654) to JYHY and PY; Discovery Early Career Researcher Award (DE170100759) and Australia National Health and Medical Research Council (NHMRC) Investigator Grant (APP1173469) to PY; Australia NHMRC Career Developmental Fellowship (APP1111338) to JYHY; Research Training Program Tuition Fee Offset and Stipend Scholarship and Chen Family Research Scholarship to YL; Australian Research Council (ARC) Postgraduate Research Scholarship and Children's Medical Research Institute Postgraduate Scholarship to HJK; University of Sydney Postgraduate Award Stipend Scholarship to YC; and NIH grant (R21DC015107) to DML. The funding source had no role in the study design; in the collection, analysis and interpretation of data; in the writing of the manuscript; and in the decision to submit the manuscript for publication.

## Author contributions

JYHY and PY conceived the study with input from YL. YL led the method development and data analysis with input from TPS, AS, PY and JYHY. YC and DML test and evaluate the method with input from HJK. JYHY, PY, YL and YC interpreted the results with input from DML, TPS, AS and HJK. YL implemented the R package with input from YC and HJK. YC implemented the Shiny app with input from JYHY and YL. JYHY, PY, TPS and YL wrote the manuscript with input from all authors. All authors read and approved the final version of the manuscript.

## Conflict of interest

The authors declare that they have no conflict of interest.

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
