## [Review Process File · Molecular Systems Biology]

scClassify: sample size estimation and multiscale classification of cells using single and multiple reference

Yingxin Lin, Yue Cao, Hani Kim, Agus Salim, Terence Speed, Dave Lin, Pengyi Yang, and Jean Yee Hwa Yang

DOI: [10.15252/msb.20199389](https://doi.org/10.15252/msb.20199389)

Corresponding author(s): Pengyi Yang (pengyi.yang@sydney.edu.au), Jean Yee Hwa Yang (jean.yang@sydney.edu.au)

Review Timeline:

Submission Date:	14th Dec 19
Editorial Decision:	12th Feb 20
Revision Received:	7th Apr 20
Editorial Decision:	20th May 20
Revision Received:	22nd May 20
Accepted:	26th May 20

Editor: Maria Polychronidou

Transaction Report:

Manuscript Number: MSB-19-9389

Title: Sample size estimation and multiscale classification of cells using single and multiple references

Thank you again for submitting your work to Molecular Systems Biology. We have now heard back from the three referees who agreed to evaluate your study. Overall, the reviewers acknowledge that while other methods exist for cell type classification of scRNA-seq data, scClassify seems to be a potentially relevant methodological contribution. They raise however a series of concerns, which we would ask you to address in a major revision.

Without repeating all the points listed below, the most substantial concerns are the following:

- The reviewers mention that the code should be provided in a form that makes it easy for the users to run and for the reviewers to evaluate it.
- Several aspects of the methodology and data analysis need to be described in better detail. Reviewer #1 provides specific recommendations related to this.
- Further information needs to be provided regarding the running times and memory requirements and how they are influenced by sample size.
- Reviewer #3 refers to the need to include comparisons to newer methods.

All other issues raised by the reviewers would need to be convincingly addressed. As you may already know, our editorial policy allows in principle a single round of major revision so it is essential to provide responses to the reviewers' comments that are as complete as possible. Please feel free to contact me in case you would like to discuss any of the issues raised by the reviewers.

REFEREE REPORTS

Reviewer #1:

Single cell RNA sequencing (scRNAseq) analysis pipelines often use an unsupervised step to

identify clusters of transcriptionally similar cell types or states. These analyses typically require time-consuming manual annotation for each cluster (for example, using known marker genes). Inspired by the increasing availability of annotated datasets or atlases, supervised approaches have been recently proposed as solution to automate this annotation. In this context, Lin et al introduce scClassify. Based on one or more reference datasets, scClassify uses the HOPACH algorithm to construct a hierarchical tree of cell types. At each level of the tree, an ensemble of kNN classifiers is trained based on multiple gene selection methods and similarity metrics. For cells that are not classified into one of the reference cell types, scClassify uses clustering for novel cell type discovery. The authors also explore experimental design aspects regarding the sample size that is required for the reference dataset(s).

I believe the idea behind scClassify is interesting, particularly in terms of using hierarchical tree of cell types as a reference. However, several aspects of the methodology require further clarification or could be improved. Detailed comments provided below.

Major comments

1. **Reproducibility.** The manuscript does not follow good reproducibility practices. As a new computational approach was introduced, it's best practice for the authors to provide the associated code or software. Moreover, as all datasets are publicly available, the authors could publish their analysis code. This will aid a more transparent benchmark comparison. For this reason, I provided a low score in terms of the quality of experimental evidence and the completeness of the supplementary information.
2. **Page 2, 2nd paragraph.** Some limitations of existing approaches are discussed. This discussion could be expanded to link with the classification shown in Supplementary Table 1. In particular, it would be good to expand Supplementary Table 1 to include the ability to use multiple reference datasets as an additional criterion. Moreover, in order to provide a better context for potential users, the authors might also consider to add additional items, such as what type of input data is required (e.g. counts, normalised counts)? what is the underlying method (e.g. kNN, GLM)? and does the method quantify uncertainty (e.g. probabilities) for the cell allocations? For examples, see related tables in [1, 2].
3. **Section 4.1.1.** The authors state that "the tree build by HOPACH algorithm is robust in general". Please clarify this sentence. Is it with respect to hyper-parameter values (e.g. the maximum number of children per branch node)? With respect to multiple reference data?
4. **Section 4.1.2.** At each branch node, 30 base classifiers are trained. These are based in a combination of 6 different similarity metrics and 5 gene selection methods. The ensemble classifier weights base classifiers according to their accuracy (similar to AdaBoost).
 - a) Figure 1c shows that some base classifiers led to very low accuracy (less than 20% in some cases). This is particularly the case for those using the DV gene selection method. This suggest that the set of base classifiers could be "pruned" to remove those with poor accuracy, reducing the computational burden of the approach.
 - b) The theory underlying AdaBoost [3] assumes that each weak classifier has higher error rate than random guessing (i.e. >50% accuracy for binary prediction). This lower bound is higher when doing multi-class classification [4], which is the relevant case for scClassify. This result could be used as a general rule to prune the set of base classifiers, removing those with low accuracy.
5. **Choice of hyper-parameters.** Please describe how were the default values for K (section 4.1.2, weighted kNN) and the correlation threshold (section 4.1.3) were chosen. How sensitive is the performance of scClassify with respect to changes in these values?
6. **Section 4.1.2, discriminative genes.** The authors rank the genes to be selected based on adjusted p-values. This can be problematic as, in most cases, the reference data was annotated by

using an unsupervised analysis (i.e. same data used twice, for clustering and for testing) [5]. Instead, the authors might consider ranking genes in terms of effect size.

7. Section 4.2. The authors mention that raw count matrices were normalised using the 'normalise' function in scater and then log-transformed.

a) This information is critical for potential users. As such, it should be included in the main text (see also my first comment).

b) I believe that the normalisation method implemented in scater corresponds to the pooling approach implemented in scran. Therefore, the reference should be [6].

c) Was a pseudo-count added prior to log-transformation?

8. Supplementary Figure 2. For these data, Garnett is not able to assign most cells. Section 4.3.1 states that Garnett requires a known list of marker genes. Is the poor performance of Garnett due to not having enough marker genes? If so, a fairer comparison could be to use one of the gene selection approaches used in scClassify (e.g. DE) to select markers.

9. Benchmarks. Recently, [1,2] performed benchmark studies in this context using (mostly) the same sources of data for the pancreas and PBMC cases. The authors should discuss how their results compare with respect to those published by [1,2]. It seems that different ranking and accuracy values were obtained. For example, for the PBMC data, [1] reports higher accuracy for scPred than scMap2Cell but the opposite is reported by Lin et al. Moreover, whilst scPred has one of the lowest accuracies in Figure 2, [2] has reported it to have high performance (F1-score) for the pancreas dataset. It would be helpful for the authors to discuss why do their results differ and potentially use additional criterion (F1 score) which could enable a direct comparison with the existing benchmarks. Finally, as the out-of-the-box SVM method was reported to have the highest overall performance by [2], I suggest it to be included as an additional comparison.

10. Section 2.5. The authors illustrate how scClassify can be applied to large scale scRNAseq datasets (e.g. Tabula Muris), but do not provide an indication of the computational burden. Please describe how does the method scale (running times) in terms of the number of cells present in the reference and query datasets.

Minor comments

1. Several figures contain axis labels that too small and hard to read (e.g. dataset names in Figure 2). Please use larger figures or font size.

2. Figure 1a. Acronyms DE, DD, P, S, etc need to be introduced.

3. Figure 3b. Please add title labels for left and right panels. Do these correspond to different experiments? Is the correlation equal to 0.98 in both cases?

4. Supplementary Figure 3. The performance of each method for each dataset is represented in terms of pie charts. While this figure does indeed summarise the overall performance of the methods, it is not straightforward to track how specific "wedges" vary across methods and datasets. The authors might consider the use of barplots instead.

References

[1] Zhao et al (2018) Briefings in Bioinformatics. PMID: 31675098

[2] Abdelaal et al (2019) Genome Biology. PMID: 31500660

[3] Freund and Schapire (1997) Journal of Computer and System Sciences.

[4] Zhu et al (2009) Statistics and its Interface

[5] Zhang et al (2019) Cell Systems. PMID: 31521605

[6] Lun et al (2016) Genome Biology. PMID: 27122128

Reviewer #2:

Here Lin et al present scClassify, a novel method for cell type classification of scRNA-seq data. Although the paper addresses an important problem in the field, there are currently several other methods available for the task. scClassify includes two interesting features that are not part of other methods, namely the ability to estimate the sample size required for a given accuracy as well as automated post-hoc clustering of unassigned cells. Both features are additions that are likely to be useful. The manuscript is well written with few typos and easy to follow. I have the following major concerns:

1 The authors compare to 11 other methods and report the accuracy. However, the comparison does not take the computational resources required into consideration. Hence, I would like to see reports of both CPU times and memory requirements for the benchmark.

2 Similarly, it would be helpful to learn how scClassify scale with sample size, both regarding time and memory. In particular, it is important to know how the size of the index scales with the number of cells and how the time to perform the mapping scales with the size of the reference.

3 I was able to download and install the R package from github. However, using it was challenging as there was no vignette available. Nevertheless, I tried a very simple test case whereby I wanted to map the Tabula Muris heart cells from FACS to the 10X heart cells. I ran the following code:

```
> heart.10x <- readRDS("data/Heart_10X.rds")
> heart.facs <- readRDS("data/Heart_FACS.rds")
> train.sc <- train_scClassify(as.matrix(counts(heart.10x)), colnames(heart.10x))
after filtering not expressed genes
[1] 15246 624
[1] "Feature Selection..."
[1] "Number of genes selected to construct HOPACH tree 10928"
[1] "Constructing tree ..."
[1] "Training..."
[1] "=====" selecting features by: limma ====="
Error in fitFDistRobustly(var, df1 = df, covariate = covariate, winsor.tail.p = winsor.tail.p) :
Variances are mostly <= 0
In addition: There were 50 or more warnings (use warnings() to see the first 50)
```

This is obviously not a very encouraging result. I got a similar error when I tried to run the scClassify() command:

```
> res.sc <- scClassify(as.matrix(counts(heart.10x)), colnames(heart.10x),
as.matrix(counts(heart.facs)), colnames(heart.facs))
Error in fitFDistRobustly(var, df1 = df, covariate = covariate, winsor.tail.p = winsor.tail.p) :
Variances are mostly <= 0
In addition: There were 50 or more warnings (use warnings() to see the first 50)
```

Although the manuscript presents compelling evidence for the performance of the method, the lack of documentation makes it difficult to use scClassify. I believe that it is vital that the users provide a suitable vignette to ensure that the method is easy to run and troubleshoot.

I have the following minor issues:

1 The color choices in fig 1d are unfortunate. It is very hard for me to distinguish pancreas from PBMC level 1

2 I may be old and curmudgeonly, but my eyes are not sharp enough to read the text on the y-axes in fig 2c. I have similar issues with the labels on fig 5c.

3 In the last paragraph on p5 the authors refer to fig 4d which I think should be 5d

Reviewer #3:

Lin et al present scClassify, a new method for classifying single cells in scRNA-seq data, as well as estimating what sample sizes would be required to reliably perform this. This is still a hard problem in scRNA-seq analysis, yet one which is important for correct interpretation of the datasets from these assays, which are rapidly producing new datasets in many systems.

Most approaches to scRNA-seq use unsupervised clustering, then manual annotation with known markers. Current approaches also ignore hierarchies. If a cell type is not in the "reference" set does not contain a particular cell type that is in the dataset that is subsequently analyzed, it tends to be forced into some inappropriate bucket.

scClassify organizes reference set cells into a hierarchy and uses multiple metrics to generate an ensemble classifier. It also permit a cell in a "new" dataset to be "unassigned rather than forcing it into one of the known cell types. The method also provides an estimate of how many cells of a given type would be needed to generate a reliable classifier, based on the reference data.

They first test a weighted kNN classifier with various similarity metrics by training on one dataset and testing on others for 6 PBMC, and 6 pancreas scRNA-seq datasets. Average accuracy is 72-93% (Figure 1c). I found the heat map in Fig 1c to be a bit uninformative since most of the cells are high (dark blue). Would it be possible to expand the color range and provide the values in a supplementary table?

When applied to the same problem, scClassify generally was better than individual methods (82% of the time), though in around 18% it was worse than any single method. This seems consistent with other studies of ensemble methods. Is there anything specific about the datasets whose performance goes down ? (e.g. sparsity, genes per cell, total UMIs, or some other measure of the dataset quality) ?

scClassify also out performed 11 other cell classifiers. However, I think several of the best performing recent ones have been missed, such as SVMreject. These are reviewed in <https://genomebiology.biomedcentral.com/articles/10.1186/s13059-019-1795-z>. In particular that review highlights variants of SVM as best. I realize that the field is moving quickly and is a moving target, but given that it has been out since last September, I think it's important to compare. That review also presents a tool for comparing performance.

The sample size estimation part of scClassify is novel (as far as I am aware) and should be useful.

Classification performance depends on the heterogeneity of each cell population, which makes sense in retrospect but I have not seen it mentioned explicitly elsewhere.

One key thing missing from the manuscript is an indication of running time. It would seem important to know how scClassify scales with the number of reference cell types, and the number of cells to be classified; and how this compares to the other methods.

Other comments and questions:

What happens when there is no intrinsic hierarchy ? e.g. in a tissue (or tumor) there will be a mixture of cell types that are not related to each other (beyond the embryonic or progenitor stage).

In Figure 4, the first sentence of the caption ends prematurely.

Summary of key concerns from editor

Response: We thank the editor for summarising the key concerns below. We have addressed each of them and also all comments from the three reviewers point-by-point below in full and to the best of our ability.

1. The reviewers mention that the code should be provided in a form that makes it easy for the users to run and for the reviewers to evaluate it.

Response: We appreciate this comment and have now provided code that was used for generating all figures as R markdown files on the public Github repository (https://github.com/SydneyBioX/scClassify_analysis). We have also submitted the associated code and software to Bioconductor (<https://github.com/Bioconductor/Contributions/issues/1446>) and tested thoroughly all examples included in our R package to make sure that they are reproducible and easy for users to run and for the reviewers to evaluate.

2. Several aspects of the methodology and data analysis need to be described in better detail. Reviewer #1 provides specific recommendations related to this.

Response: We have now included more descriptions to the methodology and data analysis in the revised manuscript according to comments from all reviewers and followed to the specific recommendations from reviewer #1. Please see our response to reviewer #1 below for details.

3. Further information needs to be provided regarding the running times and memory requirements and how they are influenced by sample size.

Response: We thank the Editor and Reviewers for this suggestion. We have now applied the same dataset (Tabula Muris) as in Abdelaal et al (PMID: 31500660) and used stratified random sampling from each cell type to evaluate the computation time of each method and memory requirements. Specifically, our extensive simulation studies can be divided into three broad categories. We (i) fixed the number of cells in query dataset as 2000 and varied the number of cells in the reference dataset (similar to the approach used in PMID: 31500660), (ii) fixed the number of cells in the reference dataset as 2000 and varied the number of cells in query dataset, (iii) fixed the number of cells in both reference and query datasets to be 5000 and varied the number of cell types in both from 4 to 12. The computation time is the sum of time spent on model training and query classification, which we believe is a fair measurement because some methods require training and query datasets to be provided together in one go. The memory requirement is the total allocated amount. The figure attached below summarises our findings.

Figure R1

In particular, for (i), our results on computation time (panel A) are highly consistent with those reported in Abdelaal et al (panel F of Fig. 7 in Abdelaal et al), and in all comparisons, scClassify (in red line) performed comparably to other methods in terms of computation time and memory usage. We have now included these new analyses in Appendix Fig S2 and revised the manuscript accordingly to discuss these new results and describe the experimental details (see section “scClassify benefits from ensemble learning and outperforms existing supervised methods”, page 5; and “Memory and running time comparison” in the Methods and Protocols section, page 18-19).

4. Reviewer #3 refers to the need to include comparisons to newer methods.

Response: We have taken on board this suggestion and have now also included in our comparison SVMreject, the overall best performing classifier in Abdelaal et al (PMID: 31500660) and CaSTLe (PMID: 30304022), a recent method based on transfer learning and a high performing one according to Abdelaal et al. As suggested by Reviewer #1, we have also included the comparison results from using Garnett with differentially expressed genes selected by scClassify. We have included these new results in the revised manuscript, section “scClassify benefits from ensemble learning and outperforms existing supervised methods” (page 4-5) and in Fig 2A-C, Fig EV 2A with further details in Appendix Fig S1, and the Methods and Protocols section “Benchmarking and method comparison” (page 17-18).

Reviewer #1:

Single cell RNA sequencing (scRNAseq) analysis pipelines often use an unsupervised step to identify clusters of transcriptionally similar cell types or states. These analyses typically require time-consuming manual annotation for each cluster (for example, using known marker genes). Inspired by the increasing availability of annotated datasets or atlases, supervised approaches have been recently proposed as solution to automate

this annotation. In this context, Lin et al introduce scClassify. Based on one or more reference datasets, scClassify uses the HOPACH algorithm to construct a hierarchical tree of cell types. At each level of the tree, an ensemble of kNN classifiers is trained based on multiple gene selection methods and similarity metrics. For cells that are not classified into one of the reference cell types, scClassify uses clustering for novel cell type discovery. The authors also explore experimental design aspects regarding the sample size that is required for the reference dataset(s).

I believe the idea behind scClassify is interesting, particularly in terms of using hierarchical tree of cell types as a reference. However, several aspects of the methodology require further clarification or could be improved. Detailed comments provided below.

Response: We thank the reviewer for the constructive feedback. We have addressed each raised point in detail below.

Major comments

1. Reproducibility. The manuscript does not follow good reproducibility practices. As a new computational approach was introduced, it's best practice for the authors to provide the associated code or software. Moreover, as all datasets are publicly available, the authors could publish their analysis code. This will aid a more transparent benchmark comparison. For this reason, I provided a low score in terms of the quality of experimental evidence and the completeness of the supplementary information.

Response: We appreciate this comment and strongly believe in reproducible research. We have submitted the associated code and software to Bioconductor and have added analysis code and scripts for reproducing all results on the scClassify Github website (https://github.com/SydneyBioX/scClassify_analysis). Please note, there are minor differences as some methods are not fully deterministic in its implementation but overall conclusions remain the same. The figures within the manuscript are a selected subset from the figures in the knit-documents, due to space limitation and ease of presentation, the multiple subfigures are manually combined in the manuscript.

2. Page 2, 2nd paragraph. Some limitations of existing approaches are discussed. This discussion could be expanded to link with the classification shown in Supplementary Table 1. In particular, it would be good to expand Supplementary Table 1 to include the ability to use multiple reference datasets as an additional criterion. Moreover, in order to provide a better context for potential users, the authors might also consider to add additional items, such as what type of input data is required (e.g. counts, normalised counts)? what is the underlying method (e.g. kNN, GLM)? and does the method quantify uncertainty (e.g. probabilities) for the cell allocations? For examples, see related tables in [1, 2].

Response: As suggested, we have added additional 4 additional columns to the original Supplementary Table 1 including:

- *Underlying methods*
- *Typical input data*
- *Ability to use multiple reference datasets*
- *Method to quantify uncertainty*

The table is now referred to as Appendix Table S1 of the revised manuscript.

3. Section 4.1.1. The authors state that "the tree build by HOPACH algorithm is robust in general". Please clarify this sentence. Is it with respect to hyper-parameter values (e.g. the maximum number of children per branch node)? With respect to multiple reference data?

Response: By that we mean (i) the scClassify is robust (or insensitive) to the choice of branch node parameter (i.e. the max number of children allowed per branch node in HOPACH algorithm) and its choice has minimum impact on scClassify classification results; and (ii) scClassify (including HOPACH component) is robust to training data perturbation. We have now performed additional experiments to demonstrate both points. These

include using different branch node parameters in HOPACH tree (included as Appendix Fig S3) and perturbing training data (i.e. 80% random subsampling of cells) (included as Fig EV2B) (both figures are attached here). We have also clarified these in the revised manuscript (section “scClassify benefits from ensemble learning and outperforms existing supervised methods”, page 5).

Ten time repeat of 80% random subsampling of training data in each pair classification pair

Figure R2

4. Section 4.1.2. At each branch node, 30 base classifiers are trained. These are based in a combination of 6 different similarity metrics and 5 gene selection methods. The ensemble classifier weights base classifiers according to their accuracy (similar to AdaBoost).

a) Figure 1c shows that some base classifiers led to very low accuracy (less than 20% in some cases). This is particularly the case for those using the DV gene selection method. This suggest that the set of base classifiers could be "pruned" to remove those with poor accuracy, reducing the computational burden of the approach.

b) The theory underlying AdaBoost [3] assumes that each weak classifier has higher error rate than random guessing (i.e. >50% accuracy for binary prediction). This lower bound is higher when doing multi-class classification [4], which is the relevant case for scClassify. This result could be used as a general rule to prune the set of base classifiers, removing those with low accuracy.

Response: We appreciate the suggestion for base classifier pruning. While we agree that removing base classifiers with low accuracy may improve ensemble classification accuracy, we note that for scClassify we have already implemented a “soft-pruning” scheme where base classifiers were weighted by their classification accuracies in the final ensemble predictions. Specifically, we note the following:

“... weights of the base classifiers are calculated as follows:

$$\alpha_t = \ln \frac{1-\epsilon_t}{\epsilon_t},$$

where ϵ_t is the error rate achieved by training and testing the base classifier on the reference dataset.”

Nevertheless, we have now updated the scClassify package to allow users to specify an accuracy cutoff for base classifier pruning if they wish to do so. A brief discussion of these details is also presented in the revised manuscript (section “Component 2: Ensemble of base classifiers” under Methods and Protocols, page 13).

5. Choice of hyper-parameters. Please describe how were the default values for K (section 4.1.2, weighted kNN) and the correlation threshold (section 4.1.3) were chosen. How sensitive is the performance of scClassify with respect to changes in these values?

Response: We understand that the choice of hyper-parameters is heuristic and mostly based on empirical results. Specifically, in current implementation of scClassify, the choice of value of k is predefined (k=10), and this value is not sensitive to the classification accuracy (See left panel of the Figure R3), and the choice of correlation threshold is determined dynamically by the normalmixEM algorithm implemented in the package of mixtools, and it is generally better than a hard coded threshold (See right panel of Figure R3).

We have now

- included these results as Fig EV2C and Fig EV2D;
- revised the manuscript to discuss these points (see section “scClassify benefits from ensemble learning and outperforms existing supervised methods”, page 5);
- included the sensitivity analysis details (see the Methods and Protocols section “Sensitivity analysis of hyperparameters of scClassify”, page 21); and
- updated the scClassify package to include these as input-parameters (thought parameter “k” and “cor_threshold_static” in scClassify()) enabling users to overwrite the default values based on their expertise and/or known characteristics of their data.

Figure R3

6. Section 4.1.2, discriminative genes. The authors rank the genes to be selected based on adjusted p-values. This can be problematic as, in most cases, the reference data was annotated by using an unsupervised analysis (i.e. same data used twice, for clustering and for testing) [5]. Instead, the authors might consider ranking genes in terms of effect size.

Response: We appreciate the point the reviewer made regarding the annotation of the reference datasets being generated by some clustering analyses and/or prior knowledge of cell types from their respective publications. However, we would like to emphasise that we are not performing any re-annotation of either the reference data or test data (by clustering or any other methods) and therefore did not use the “same data twice” for testing.

We note that all supervised methods assume cell type annotations in the reference data are mostly correct and therefore can be used for training an informative classification model. Using the annotation in the

reference data for ranking and selecting genes follows the same assumption of supervised learning. The ranking of genes by p-values accounting for the sign (up and down regulation) would be the same as ranking by the corresponding effect size (t-statistics). Depending on the different multiple testing adjustment method, the ranking of genes by adjusted p-value will have a similar ranking.

7. Section 4.2. The authors mention that raw count matrices were normalised using the 'normalise' function in scater and then log-transformed.

a) This information is critical for potential users. As such, it should be included in the main text (see also my first comment).

b) I believe that the normalisation method implemented in scater corresponds to the pooling approach implemented in scan. Therefore, the reference should be [6].

c) Was a pseudo-count added prior to log-transformation?

Response: We agree with the reviewer regarding (a) and have now included the details of normalisation and transformation in the revised manuscript (see section Introduction, page 3). For (b) and (c), we used the default setting in the "normalize" function of scater (version 1.13.7). This function first computes the normalized expression values by dividing the counts for each cell by the size factor for that cell, and then log-normalized values are calculated by adding the pseudo-count (default is 1) to the normalized count and performing a log2 transformation. The size factor, by default, uses the function "librarySizeFactors", which is defined as the per-cell size factors from the library sizes as the total sum of counts per cell. These details are included in the revised manuscript (see the Methods and Protocols section "Data collections and processing", Page 16). Following comment #1, all analytical code is now made available at https://github.com/SydneyBioX/scClassify_analysis.

8. Supplementary Figure 2. For these data, Garnett is not able to assign most cells. Section 4.3.1 states that Garnett requires a known list of marker genes. Is the poor performance of Garnett due to not having enough marker genes? If so, a fairer comparison could be to use one of the gene selection approaches used in scClassify (e.g. DE) to select markers.

Response: We note that for Garnett, we used the marker file provided on the Author's website (<https://cole-trapnell-lab.github.io/garnett/classifiers/>) and those published in literature (Pang et al, 2005; Collin & Bigley, 2018; Scarlett et al, 2011; Lawlor et al, 2017; Muraro et al, 2016). These details are included in the Methods and Protocols section "Benchmarking and method comparison", page 17 -18. As suggested, we have now also performed Garnett using the DE genes selected in scClassify. These results are included in Fig 2A-C and Fig EV 2A and we have revised the manuscript accordingly (page 17-18).

9. Benchmarks. Recently, [1,2] performed benchmark studies in this context using (mostly) the same sources of data for the pancreas and PBMC cases. The authors should discuss how their results compare with respect to those published by [1,2]. It seems that different ranking and accuracy values were obtained. For example, for the PBMC data, [1] reports higher accuracy for scPred than scMap2Cell but the opposite is reported by Lin et al. Moreover, whilst scPred has one of the lowest accuracies in Figure 2, [2] has reported it to have high performance (F1-score) for the pancreas dataset.

Response: We note that Zhao et al (PMID: 31675098) used traditional binary classification evaluation metrics based on true positive, true negative, false positive, and false negative. Since our evaluation metric, extended on de Kanter et al (PMID: 31226206), also includes categories such as "incorrectly unassigned", "incorrectly assigned", and "correctly unassigned", it is not unexpected that we would have differences in ranking of methods compared to those from Zhao et al. As suggested, we have now included a brief discussion of these points in the "Discussion" section of the revised manuscript (page 9).

In particular, for the pancreas data classification, Abdelaal et al (PMID: 31500660) used both intra-data classification (i.e. training and test within the same scRNA-seq dataset) and inter-data classification (i.e. training on one dataset and test on the others). The latter is similar to our evaluation framework. However, in Abdelaal et al only four major endocrine pancreatic cell types (alpha, beta, delta, and gamma) were selected

from each of the four datasets (Baron, Muraro, Segerstolpe and Xin) for benchmark classification (see Figure 5 and methods on inter-data classification in Pancreas section of Abdelaal et al). In comparison, in our experiment we included two additional pancreatic datasets (Lawlor and Wang) besides the four datasets used in Abdelaal et al. Moreover, we included almost all cell types that are reported in each of the original papers (see “Data collections and processing” under the Methods and Protocols section, page 16). The additional datasets and cell types included in our study have created more challenging scenarios such as imbalanced and rare classes and unrepresented cell types in the training and test data pairs, which together with the difference in evaluation metric have led to the difference in the benchmarking results.

It would be helpful for the authors to discuss why do their results differ and potentially use additional criterion (F1 score) which could enable a direct comparison with the existing benchmarks.

Response: We believe the nature of multi-class cell type classification is more complex than a typical binary classification problem. Here, we extended on the evaluation framework introduced by de Kanter et al (PMID: 31226206), which considers categories such “incorrectly unassigned”, “incorrectly assigned”, and several other categories and a classical F1-score based on precision and recall alone, does not capture such complexity. As such it is not an informative evaluation metric for our benchmarking study. As suggested, we have now discussed this in the revised manuscript (page 9).

- Finally, as the out-of-the-box SVM method was reported to have the highest overall performance by [2], I suggest it to be included as an additional comparison.

Response: As suggested, we have now included SVMreject (SVM that allows unassignment), which was the overall best performing classifier according to Abdelaal et al (PMID: 31500660), in our comparison. These additional results are included in revised Fig 2A-C and Fig EV2A.

10. Section 2.5. The authors illustrate how scClassify can be applied to large scale scRNAseq datasets (e.g. Tabula Muris), but do not provide an indication of the computational burden. Please describe how does the method scale (running times) in terms of the number of cells present in the reference and query datasets.

Response: We thank the reviewer for this suggestion. Please see our response to Editor’s comment 3 for details. Specifically, panels A and B address this comment on method scalability to the number of cells present in the reference and query datasets. Overall, our results are highly consistent with those reported in Abdelaal et al (panel F of Fig. 7 in Abdelaal et al) and scClassify performed comparably to other methods in terms of computation time.

Minor comments

1. Several figures contain axis labels that too small and hard to read (e.g. dataset names in Figure 2). Please use larger figures or font size.

Response: As suggested, we have now increased the font size in these panels (including Figure 2).

2. Figure 1a. Acronyms DE, DD, P, S, etc need to be introduced.

Response: As suggested, we have now included the full name of each abbreviation in the legend of Figure 1A.

3. Figure 3b. Please add title labels for left and right panels. Do these correspond to different experiments? Is the correlation equal to 0.98 in both cases?

Response: As suggested, we have now included the title labels for left and right panels of Figure 3B. The correlation for the right panels (PBMC level 2) is 0.99. Thank you for identifying this typo.

4. Supplementary Figure 3. The performance of each method for each dataset is represented in terms of pie charts. While this figure does indeed summarise the overall performance of the methods, it is not straightforward to track how specific "wedges" vary across methods and datasets. The authors might consider the use of barplots instead.

Response: As suggested, we have now created barplots for these results and included them in Fig EV2 of the revised manuscript. We have also included individual barplots for each evaluation categories (illustrated as Figure R4 below), however, as not all methods allow intermediate and unassigned, the individual barplots didn't increase clarity.

Figure R4

References

- [1] Zhao et al (2018) Briefings in Bioinformatics. PMID: 31675098
- [2] Abdelaal et al (2019) Genome Biology. PMID: 31500660
- [3] Freund and Schapire (1997) Journal of Computer and System Sciences.
- [4] Zhu et al (2009) Statistics and its Interface

- [5] Zhang et al (2019) Cell Systems. PMID: 31521605
[6] Lun et al (2016) Genome Biology. PMID: 27122128

Reviewer #2:

Here Lin et al present scClassify, a novel method for cell type classification of scRNA-seq data. Although the paper addresses an important problem in the field, there are currently several other methods available for the task. scClassify includes two interesting features that are not part of other methods, namely the ability to estimate the sample size required for a given accuracy as well as automated post-hoc clustering of unassigned cells. Both features are additions that are likely to be useful. The manuscript is well written with few typos and easy to follow. I have the following major concerns:

Response: We thank the reviewer for highlighting the novel aspects and expected utilities of scClassify. We have addressed the comments raised by the reviewer point by point below.

1 The authors compare to 11 other methods and report the accuracy. However, the comparison does not take the computational resources required into consideration. Hence, I would like to see reports of both CPU times and memory requirements for the benchmark.

Response: Thank you for suggesting these evaluation criteria. Note that this is discussed in detail in response to the Editor's comment 3. In brief, we have now included the assessment of computation time and memory requirements for each method when applied to Tabula Muris dataset with varying numbers of cells (ranging from 100 to 30,000). We found that, overall, scClassify is comparable to other methods in terms of computational resources required for large data classification. We have now included these new results in the revised manuscript in section "scClassify benefits from ensemble learning and outperforms existing supervised methods" in page 4 and section "Memory and running time comparison" in Methods and Protocols (page 18-19).

2 Similarly, it would be helpful to learn how scClassify scale with sample size, both regarding time and memory. In particular, it is important to know how the size of the index scales with the number of cells and how the time to perform the mapping scales with the size of the reference.

Response: This is related to Comment 1 above, please see our response to Editor's comment 3 regarding computation time and memory requirement for each method.

3 I was able to download and install the R package from github. However, using it was challenging as there was no vignette available. Nevertheless, I tried a very simple test case whereby I wanted to map the Tabula Muris heart cells from FACS to the 10X heart cells. I ran the following code:

```
> heart.10x <- readRDS("data/Heart_10X.rds")
> heart.facs <- readRDS("data/Heart_FACS.rds")
> train.sc <- train_scClassify(as.matrix(counts(heart.10x)), colnames(heart.10x))
after filtering not expressed genes
[1] 15246 624
[1] "Feature Selection..."
[1] "Number of genes selected to construct HOPACH tree 10928"
[1] "Constructing tree ..."
[1] "Training...."
[1] "===== selecting features by: limma ====="
Error in fitFDistRobustly(var, df1 = df, covariate = covariate, winsor.tail.p = winsor.tail.p) :
Variances are mostly <= 0
In addition: There were 50 or more warnings (use warnings() to see the first 50)
```

This is obviously not a very encouraging result. I got a similar error when I tried to run the `scClassify()` command:

```
> res.sc <- scClassify(as.matrix(counts(heart.10x)), colnames(heart.10x), as.matrix(counts(heart.facs)),
colnames(heart.facs))
Error in fitFDistRobustly(var, df1 = df, covariate = covariate, winsor.tail.p = winsor.tail.p) :
Variances are mostly <= 0
In addition: There were 50 or more warnings (use warnings() to see the first 50)
```

Although the manuscript presents compelling evidence for the performance of the method, the lack of documentation makes it difficult to use `scClassify`. I believe that it is vital that the users provide a suitable vignette to ensure that the method is easy to run and troubleshoot.

Response: We appreciate this comment. The above returned errors are due to the following two reasons:

- 1. cellTypes_train input is the column names of the training data instead of the cell type information of the training data.*
- 2. The input of the expression matrix is count data instead of log-transformed normalised matrix.*

We have now documented the functions in scClassify thoroughly and added detailed examples for each function. A detailed Vignette with an illustrated example using Pancreas datasets is provided at: (<https://sydneybioinformatics.github.io/scClassify/articles/scClassify.html>). Additional examples with larger datasets such as training on Tabula Muris Heart cells are also available on github site: (https://sydneybioinformatics.github.io/scClassify/articles/webOnly/scClassify_TM_heart.html).

Additional tutorials about utilising pre-trained scClassify models and sample size calculation are also provided at: (<https://sydneybioinformatics.github.io/scClassify/articles/pretrainedModel.html>) and (<https://sydneybioinformatics.github.io/scClassify/articles/webOnly/sampleSizeCal.html>).

We have now thoroughly tested our package and it has been submitted to Bioconductor. We hope that the integration with other Bioconductor packages will streamline the usage of scClassify for users.

I have the following minor issues:

1 The color choices in fig 1d are unfortunate. It is very hard for me to distinguish pancreas from PBMC level 1

Response: We have updated Fig 1D by using colours that better distinguish the data points of pancreas from PBMC level 1.

2 I may be old and curmudgeonly, but my eyes are not sharp enough to read the text on the y-axes in fig 2c. I have similar issues with the labels on fig 5c.

Response: Thank you for this and we have now increased the font size for Fig 2C and Fig 5C.

3 In the last paragraph on p5 the authors refer to fig 4d which I think should be 5d

Response: We have corrected this error in the revised manuscript.

Reviewer #3:

Lin et al present `scClassify`, a new method for classifying single cells in scRNA-seq data, as well as estimating what sample sizes would be required to reliably perform this. This is still a hard problem in scRNA-seq analysis, yet one which is important for correct interpretation of the datasets from these assays, which are rapidly producing new datasets in many systems.

Most approaches to scRNA-seq use unsupervised clustering, then manual annotation with known markers. Current approaches also ignore hierarchies. If a cell type is not in the "reference" set does not contain a particular cell type that is in the dataset that is subsequently analyzed, it tends to be forced into some inappropriate bucket.

scClassify organizes reference set cells into a hierarchy and uses multiple metrics to generate an ensemble classifier. It also permit a cell in a "new" dataset to be "unassigned rather than forcing it into one of the known cell types. The method also provides an estimate of how many cells of a given type would be needed to generate a reliable classifier, based on the reference data.

Response: We thank this reviewer for summarising the positive aspects of scClassify in the above paragraphs. We have addressed all questions and comments point-by-point below.

1. They first test a weighted kNN classifier with various similarity metrics by training on one dataset and testing on others for 6 PBMC, and 6 pancreas scRNA-seq datasets. Average accuracy is 72-93% (Figure 1c). I found the heat map in Fig 1c to be a bit uninformative since most of the cells are high (dark blue). Would it be possible to expand the color range and provide the values in a supplementary table?

Response: We thank the reviewer for pointing out the average accuracy range. We have now updated the colour gradient and expanded the colour range in Figure 1C to focus on the accuracy from 50% to 100%. As suggested, we have also now provided the values in the new Dataset EV1.

2. When applied to the same problem, scClassify generally was better than individual methods (82% of the time), though in around 18% it was worse than any single method. This seems consistent with other studies of ensemble methods. Is there anything specific about the datasets whose performance goes down ? (e.g. sparsity, genes per cell, total UMIs, or some other measure of the dataset quality) ?

Response: As suggested, we have partitioned the prediction cases into (1) the ensemble model is better than the single best model and (2) a single model is better than the ensemble, and looked at a variety of data characteristics in both training and test sets including data sparsity, number of UMI, number of genes, mitochondrial gene percentage, number of cell types, the largest cell type percentage (max_pct_cellType), the smallest cell type percentage (min_pct_cellType), and mean cell type percentage (mean_pct_cellType). We also included characteristics such as the number of common cell types between the training and test data (commonCellType), number of cell types that are unique in the training data (numUnique_train), number of cell types that are unique in the test data (numUnique_test), percentage of cells that are of unique cell types in training data (pctUnique_train). Results in boxplots are included below and we didn't find any clear difference with respect to the tested data characteristics between the two groups (Figure R5).

Figure R5

- scClassify also out performed 11 other cell classifiers. However, I think several of the best performing recent ones have been missed, such as SVMreject. These are reviewed in <https://genomebiology.biomedcentral.com/articles/10.1186/s13059-019-1795-z>. In particular that review highlights variants of SVM as best. I realize that the field is moving quickly and is a moving target, but given that it has been out since last September, I think it's important to compare. That review also presents a tool for comparing performance.

Response: As suggested, we have now included SVMreject (a variant of SVM) which was reported as overall the best performing method in Abdelaal et al (PMID: 31500660) and CaSTLe (PMID: 30304022) (a recent method based on transfer learning) which is ranked relatively high but not included in our initial comparison. All other highly ranked methods in Abdelaal et al are already included in our initial comparison. We have updated Fig 2A-C, Fig EV2, and Appendix Fig S1 and revised the manuscript accordingly (see the Methods and Protocol section "Benchmarking and method comparison", page 17-18) to include these results.

4. The sample size estimation part of scClassify is novel (as far as I am aware) and should be useful.

Response: We thank the reviewer for this positive comment on the novelty and the utility of our proposed sample size estimation.

5. Classification performance depends on the heterogeneity of each cell population, which makes sense in retrospect but I have not seen it mentioned explicitly elsewhere.

Response: The more heterogeneity the cell types are in the reference data, the easier they can be distinguished by the classification algorithm. In our sample size simulation study, Appendix Figure S4 shows that the classification performance is impacted by within-population heterogeneity. This point is now discussed in the "Discussion" section of the revised manuscript (page 9).

6. One key thing missing from the manuscript is an indication of running time. It would seem important to know how scClassify scales with the number of reference cell types, and the number of cells to be classified; and how this compares to the other methods.

Response: Thanks for these suggestions. This is a similar comment to Reviewers #1 and #2 and we have addressed this together in response to Editor's comment 3. Please refer to panels A and C that are specifically related to the scalability of scClassify with the number of cells and number of cell types in the reference data. Overall, we found scClassify to be comparable to other methods and these findings have been included in the manuscript accordingly.

Other comments and questions:

7. What happens when there is no intrinsic hierarchy ? e.g. in a tissue (or tumor) there will be a mixture of cell types that are not related to each other (beyond the embryonic or progenitor stage).

Response: In such a case, cell types may not have a biological hierarchy but will be related to each other by the similarity of their transcriptome profiles which is still a useful organisation of reference data for classification.

8. In Figure 4, the first sentence of the caption ends prematurely.

Response: We have corrected this in the revised manuscript.

Manuscript Number: MSB-19-9389R, Sample size estimation and multiscale classification of cells using single and multiple references

Thank you for sending us your revised manuscript. We have now heard back from the three reviewers who were asked to evaluate your study. As you will see below, the reviewers are satisfied with the modifications made and are supportive of publication.

Before we formally accept your manuscript for publication we would ask you to address some remaining editorial issues.

REFeree REPORTS

Reviewer #1:

The authors have addressed all my comments.

Reviewer #2:

The authors have addressed all of my comments, in particular they have added a very nice-looking vignette.

Reviewer #3:

First, my apologies to the authors and editor for the delay in responding to the revised manuscript. Strange times are upon us.

The authors have addressed my comments, and those of the other reviewers. Having recently struggled to get decent results from alternative algorithms, we look forward to trying out scClassify more intensively in the wild. This will be facilitated by its inclusion in Bioconductor. The vignettes are useful and clear. I was not able to access the Shiny app at the moment, but that provision should make it usable by researchers who are not experienced with R.

The Authors have made the requested editorial changes.

Manuscript number: MSB-19-9389RR, scClassify: sample size estimation and multiscale classification of cells using single and multiple reference

Thank you again for sending us your revised manuscript and for performing the requested changes. We are now satisfied with the modifications made and I am pleased to inform you that your paper has been accepted for publication.

Corresponding Author Name: Jean Yee Hwa Yang, Pengyi Yang

Manuscript Number: MSB-19-9389